# Murine T-cell receptor OT-I exhibits imperfect discrimination between foreign and self-antigens

Anna Huhn[1], Mikhail A Kutuzov [ID][1], Keir Maclean[1], Lion F K Uhl [ID][2,3], Jagdish M Mahale[2], Audrey Gérard [ID][2], P Anton van der Merwe [ID][1] & Omer Dushek [ID][1][✉]

## Abstract

T cells use their T-cell receptors (TCRs) to discriminate between higher-affinity foreign and lower-affinity self-peptide-MHC (pMHC) antigen complexes. The OT-I mouse TCR is widely used to study antigen discrimination between foreign and self-pMHC antigens, and previous work suggested it achieved near-perfect discrimination between higher- and lower-affinity antigens. However, other TCRs show imperfect discrimination. To resolve these discrepancies, we developed in this study a protocol for measuring ultra-low TCR-pMHC binding affinities to determine the 3D solution affinities of OT-I TCR for 19 pMHCs. These revised 3D affinities now strongly correlate with 2D membrane affinities and predict T-cell functional responses. Our results indicate that OT-I exhibits enhanced yet imperfect discrimination, similar to other TCRs, explaining how T cells can detect abnormally high levels of low-affinity self-antigens. We also show that OT-I discrimination is consistent with the kinetic proofreading model, which highlights that discrimination is most effective for low-affinity pMHC ligands. Our work underscores the ability of T cells to gauge proxies for 3D affinity within the 2D interface, with implications for the mechanisms underlying antigen discrimination.

**Keywords** T cell; Antigen Discrimination; OT-I TCR; Affinity; Surface Plasmon Resonance
**Subject Categories** Immunology; Structural Biology

See also: P Robert & P Bongrand

## Introduction

T cells orchestrate adaptive immune responses by recognising infected or cancerous cells whilst ignoring normal cells. This ability relies on their T-cell receptors (TCRs) being able to discriminate between higher-affinity foreign and lower-affinity self-peptide antigens presented on major histocompatibility complexes (pMHCs) on antigen presenting cell (APC) surfaces. This process

was first explored using mouse T-cell hybridomas and transgenic mice expressing TCRs of defined specificity (Alam et al, 1999, 1996; Hogquist et al, 1995; Kersh et al, 1998; Kersh and Allen, 1996; Lyons et al, 1996; Rosette et al, 2001). The OT-I TCR transgenic mouse has been among the most widely used in vivo systems for studying T-cell responses, including central and peripheral tolerance, infection, cancer, vaccination, autoimmunity, and transplantation (Daniels et al, 2006; Drobek et al, 2018; Juang et al, 2010; Ma et al, 2020; Mazet et al, 2023; Navarro et al, 2014; Preston et al, 2015; Shimizu et al, 2021; Stepanek et al, 2014; Uhl et al, 2023; Wilson et al, 2009). The OT-I TCR, which recognises the ovalbumin-derived peptide SIINFEKL (N4) presented by the MHC class I protein H-2K$^b$, has also been used to investigate the molecular and cellular mechanisms underlying T cell antigen recognition (Altan-Bonnet and Germain, 2005; Francois et al, 2013; Huang et al, 2010; Jiang et al, 2011; Liu et al, 2014; Lo et al, 2023, 2018).

During the past three decades, more than 20 additional peptides have been used to investigate how varying the peptide affects antigen recognition by OT-I T cells. Early studies suggested that the OT-I TCR exhibits near-perfect antigen discrimination (Alam et al, 1999, 1996; Altan-Bonnet and Germain, 2005; Hogquist et al, 1995; Rosette et al, 2001). For example, while the OT-I TCR was reported to bind the E1 peptide with only a 3-fold lower affinity than the N4 peptide, it required a 100,000-fold higher concentration to be activated (Alam et al, 1999). This striking observation spurred extensive theoretical and experimental efforts to uncover the mechanism(s) enabling such exceptional discrimination (Aleksic et al, 2010; Altan-Bonnet and Germain, 2005; Choi et al, 2023; Dushek et al, 2009; Dushek and van der Merwe, 2014; Fernandes et al, 2019; Francois et al, 2013; Ganti et al, 2020; Govern et al, 2010; Lever et al, 2014; Prüstel and Meier-Schellersheim, 2024; Robert et al, 2012; Schamel et al, 2005). It has been proposed that solution or 3D binding properties (e.g., affinities, on-rates and off-rates) between soluble forms of TCRs and pMHCs, commonly measured using surface plasmon resonance (SPR), may not correlate with the 2D binding properties measured between membrane-attached TCRs and pMHCs (Van der Merwe, 2001). Differences between 3D and 2D binding properties can be a consequence of pulling or pushing forces on the TCR/pMHC bond. These include microvilli-like protrusions that push the T-cell membrane into the target cell (Cai et al, 2017; Sage et al, 2012) and

[1]Sir William Dunn School of Pathology, University of Oxford, Oxford, UK. [2]Kennedy Institute of Rheumatology, University of Oxford, Oxford, UK. [3]Present address: Memorial Sloan Kettering Cancer Center, New York, NY, USA. [✉]E-mail: omer.dushek@path.ox.ac.uk

large surface molecules that push membranes apart (Allard et al, 2012; van der Merwe and Dushek, 2011). Indeed, 2D affinity measurements of OT-I/pMHC interactions revealed much larger variations, with a 200-fold difference between the N4 and E1, compared with only 3-fold variation in 3D affinity (Huang et al, 2010). Thus, the OT-I TCR apparently displays a highly non-linear relationship between 3D and 2D affinities.

However, findings with the OT-I TCR have not been replicated with other TCRs. Using an optimised SPR protocol for measuring very low-affinity TCR/pMHC interactions, we have shown that the 1G4 and A6 human TCRs display much weaker discrimination than originally reported for the OT-I TCR, in that a threefold lower peptide affinity requires only a ninefold increase in peptide concentration to activate T cells (Pettmann et al, 2021). We also performed a meta-analysis of the published literature, confirming the same result with other human and mouse TCRs (Pettmann et al, 2021). Furthermore, unlike the OT-I TCR, the 3D and 2D affinities of the 1E6 produced linear correlations (Cole et al, 2016), while the 3D and 2D TCR/pMHC lifetimes for the 5 c.c7 TCR were comparable (O'Donoghue et al, 2013). These discrepancies between the OT-I TCR and other TCRs remain unexplained.

One advantage of the OT-I TCR is that some of the self-peptides that this TCR binds (e.g., Catnb and Cappa1) have been identified, based on their ability to positively select OT-I thymocytes (Santori et al, 2002). Given that mature OT-I T cells must ignore these self-peptides, measuring the affinities of OT-I binding these peptides would provide insights into the level of TCR discrimination required for tolerance. Unfortunately, it has not been possible to measure these affinities at physiological temperatures; $K_D$ estimates have only been reported at 10 °C (Juang et al, 2010).

Concentrating soluble proteins to the levels required for studying very low-affinity interactions by SPR often results in the formation of protein aggregates, which bind with slow kinetics (Davis et al, 1998; Van Der Merwe and Barclay, 1996). In early SPR studies, the OT-I TCR showed biphasic binding to N4 pMHC, with one component binding with unusually slow kinetics ($k_{on}$ ~0.03 μM$^{-1}$ s$^{-1}$, $k_{off}$ ~ 0.02 s$^{-1}$ for the slow phase) (Alam et al, 1999). This gave rise to the notion that the OT-I TCR interaction with N4 has unusually high affinity. However, this slow phase is also consistent with the presence of OT-I TCR aggregates. This is supported by subsequent studies that reported monophasic binding with much faster kinetics to N4 (Liu et al, 2015; Pettmann et al, 2023; Stepanek et al, 2014). Collectively, this suggests that inaccurate 3D affinity measurements could explain the apparent discrepancies between the OT-I TCR and other TCRs.

These discrepancies and lack of affinity data for most OT-I peptides motivated us to use our optimised SPR protocol to measure 3D affinities between the OT-I TCR and 20 commonly used peptides at 37 °C (Pettmann et al, 2021). We now report that the OT-I TCR binds N4 with physiological affinity and displays much wider differences in affinity for various peptides, such as a 100-fold lower affinity for E1 relative to N4 rather than the originally reported threefold difference. Our revised 3D $K_D$ values correlate with 2D $K_D$ values, and indicate that the discriminatory power of the OT-I TCR is comparable to other TCRs, and increases for lower-affinity antigens. These findings reconcile apparent discrepancies between the OT-I TCR and other TCRs, and have important implications for understanding TCR antigen discrimination.

# Results

## Systematic measurements of OT-I TCR affinities at 37 °C

We selected a panel of 20 peptides commonly used in the literature for OT-I TCR experiments, including positively selecting self-peptides (Table 1). Purified OT-I TCR was injected over surfaces immobilised with each pMHC at 37 °C (Fig. 1). The association and dissociation phases were too fast to allow rate constants to be estimated, and with the exception of the N4 pMHC, binding did not saturate at the OT-I TCR concentration range tested. This is consistent with weak interactions. As a result, the $K_D$ could not be accurately determined by conventional fitting of the 1:1 binding model where the maximal TCR binding ($B_{max}$) is unconstrained.

To avoid concentrating the TCR, which can introduce protein aggregates, we used a previously described SPR protocol that does not require TCR binding to approach saturation (Pettmann et al, 2021). We first determined the maximum TCR binding ($B_{max}$) for surfaces immobilised with different levels of the high-affinity N4 pMHC, where TCR binding saturates at attainable concentrations of TCR (Fig. 1A,B). We then injected the pMHC-specific B2M or Y3 antibody, enabling us to produce a standard curve that relates antibody binding to the TCR $B_{max}$ (Fig. 1B). Both antibodies produced an identical standard curve, indicating that the detection of correctly folded pMHC does not depend on the antibody clone. For low-affinity TCR/pMHC interactions, where TCR binding does

**Table 1. Revised $K_D$ values for OT-I specific peptides at 37 °C.**

| Peptide | Sequence | $K_D$ apparent[a] | | $K_D$ active[b] | | |
| --- | --- | --- | --- | --- | --- | --- |
| | | Mean[c] | SD[c] | Mean[c] | SD[c] | N |
| N4 | SIINFEKL | 33.97 | 1.135 | 34.46 | 1.189 | 24 |
| A2 | SAINFEKL | 91.44 | 1.095 | 97.5 | 1.084 | 4 |
| Q4 | SIIQFEKL | 188.8 | 1.231 | 194.3 | 1.235 | 5 |
| Q4R7 | SIIQFERL | 239.9 | 1.313 | 252 | 1.325 | 3 |
| T4 | SIITFEKL | 344.4 | 1.151 | 388.7 | 1.166 | 4 |
| Q7 | SIINFEQL | 481.1 | 1.264 | 523.7 | 1.284 | 3 |
| Q4H7 | SIIQFEHL | 516.6 | 1.23 | 556.9 | 1.245 | 3 |
| V4 | SIIVFEKL | 648.5 | 1.153 | 794.1 | 1.183 | 4 |
| G4 | SIIGFEKL | 680.7 | 1.083 | 941.8 | 1.114 | 4 |
| K4 | SIIKFEKL | 2053 | 1.171 | 2694 | 1.219 | 3 |
| VSV | RGYVYQGL | 1582 | 1.178 | 3103 | 1.357 | 6 |
| E4 | SIIEFEKL | 2204 | 1.139 | 3331 | 1.208 | 3 |
| Null | SIAAFASL | 1364 | 1.163 | 3361 | 1.403 | 3 |
| E1 | EIINFEKL | 999.8 | 1.243 | 3637 | 1.809 | 9 |
| R4 | SIIRFEKL | 2265 | 1.131 | 5066 | 1.313 | 6 |
| Catnb | RTYTYEKL | 1477 | 1.066 | 2024 | 1.088 | 6 |
| Cappa1 | ISFKFDHL | 1683 | 1.275 | 3956 | 1.789 | 6 |
| Mapk1 | VGPRYTNL | 1764 | 1.345 | 2335 | 1.49 | 4 |
| Stat3 | ATLVFHNL | 1916 | 1.252 | 3435 | 1.451 | 3 |
| V-OVA | RGYNYEKL | nd | nd | nd | nd | |

[a]The apparent $K_D$ (μM) of OT-I binding to both active and inactive pMHC.
[b]The $K_D$ (μM) of OT-I binding only active pMHC. [c]Geometric mean and geometric SD.

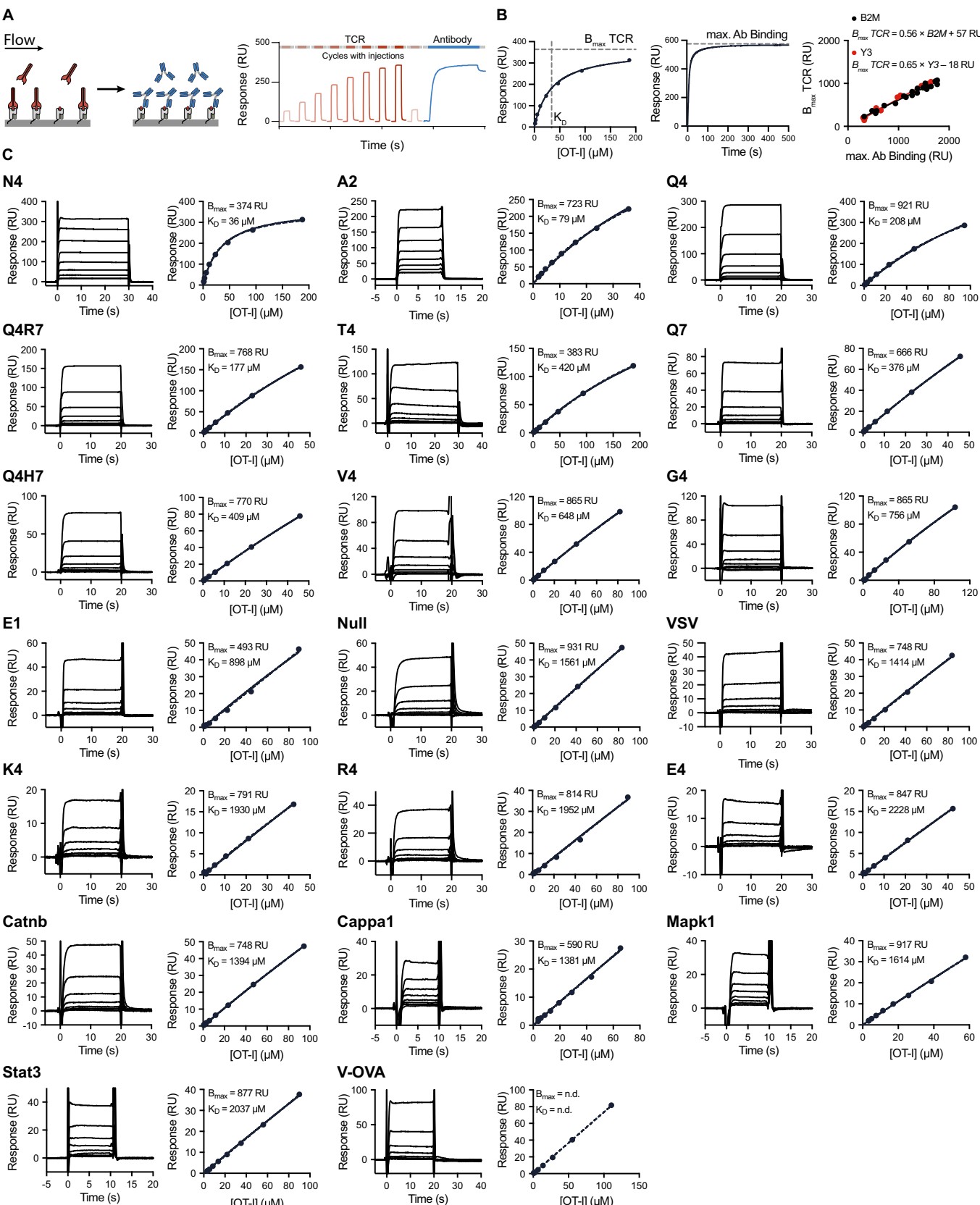

**Figure 1.   The OT-I TCR interaction with 20 commonly used peptides exhibits fast kinetics and low affinity at 37 °C.**

(A) Schematic of the SPR protocol. The TCR analyte is injected at 8 different concentrations over a surface coupled with purified pMHC, followed by an injection of a pMHC conformationally sensitive antibody (Y3 or B2M). (B) Steady-state binding response for the higher-affinity N4 pMHC fitted with a one-site specific binding model to determine TCR $B_{max}$ and $K_D$ (left). Representative sensorgram of the B2M antibody specific for human $\beta 2m$ domain injected after the final TCR injection to obtain maximum antibody binding (centre). Empirical standard curve relating maximum antibody binding (x axis) to fitted TCR $B_{max}$ obtained from N4 pMHC across $N = 23$ (B2M) and $N = 14$ (Y3) independent experiments with different levels of pMHC on the surface (right). (C) Representative SPR traces (left) and steady-state binding plots (right) for the indicated peptide. Steady state data were fitted with a 1:1 binding model with constrained $B_{max}$ to obtain $K_D$. Source data are available online for this figure.

not saturate, measuring the B2M or Y3 antibody binding after each experimental run enabled us to use this standard curve to determine the TCR $B_{max}$. This in turn allowed us to estimate the $K_D$ by fitting the usual 1:1 binding model while constraining the value of $B_{max}$. Using this protocol, we measured the apparent affinities for all OT-I peptides, which ranged from a $K_D$ of 34 µM to over 2200 µM (Figs. 1C and EV1; Table 1). We confirmed that pMHC produced in bacterial (E. coli) and mammalian (HEK293) cells produced the same affinities (Fig. EV2).

Because the pMHC antibody displayed only modest binding to V-OVA [<30 RU versus >600 RU for N4 (Fig. 2A)], we were unable to determine a $K_D$ for the OT-I TCR binding this peptide. This lack of binding of a conformationally sensitive antibody indicated that the V-OVA pMHC is not correctly folded, and suggested that the limited OT-I binding to V-OVA pMHC may be non-specific (Fig. 2A). To explore this, we tested OT-I binding to N4 pMHC after the latter had been denatured by exposure to low-pH glycine solution. This resulted in a 100-fold reduction in binding of the B2M antibody, confirming denaturation (Fig. 2A), yet the OT-I TCR continued to display up to 60 RU of binding to denatured N4 (compared to 400 RU to correctly folded N4). This suggests that incorrectly folded pMHC on the sensor surface can non-specifically bind injected analytes, including the OT-I TCR. In support of this, a control protein, Ovalbumin, also showed binding to immobilised pMHCs but not to another immobilised protein, CD86 (Appendix Fig. S1A,B). Thus, some OT-I TCR binding detected by SPR represents binding to inactive/unfolded pMHC. This non-specific binding needs to be taken into account in order to accurately measure the affinities of OT-I TCR for specific, or active, pMHCs. While the pMHC can exist in multiple conformations (Wieczorek et al, 2017; Wu et al, 2019), for our purposes, we have simplified these into canonical conformations that can bind both the TCR and conformationally sensitive antibodies, and those conformations that cannot.

To estimate the OT-I affinity for inactive V-OVA and denatured N4, we fit the steady-state data with the usual 1:1 binding model but constrained the $B_{max}$ to the total amount of pMHC immobilised (Fig. 2A, bottom). If we assume that almost all the immobilised V-OVA and denatured N4 is inactive, the immobilisation level can serve as a proxy for total available non-specific binding ($B_{max}$) because the molecular weights of pMHC (49 kDa) and OT-I TCR (51 kDa) are nearly identical. Using this method, we found that the OT-I TCR bound denatured N4 and V-OVA with $K_D$ values of 2180 and 2090 µM, respectively (Fig. 2B). While this very weak binding is unlikely to affect $K_D$ estimates when the fraction of inactive pMHC is very low, it would be expected to have an impact when the fraction of inactive pMHC is large (Fig. 2C).

We next estimated the fraction of inactive pMHC in the pMHC preparations. In the case of the high-affinity N4 pMHC, we can

directly estimate this fraction by comparing the TCR $B_{max}$ to the amount of immobilised N4 pMHC, which includes both active and inactive pMHC (Fig. 2D). If all pMHCs were active, $B_{max}$ and pMHC immobilisation levels should match, producing a slope of 1.0. Instead, the slope of the $B_{max}$ vs pMHC plot was 0.74, indicating that 74% of N4 is active (Fig. 2D,E). For lower-affinity pMHCs the binding of conformationally sensitive antibodies was used to estimate the $B_{max}$. The ratio of $B_{max}$ to pMHC immobilisation indicated that the amount of active pMHC varied from 50% (E1) to 80% (Q4) (Fig. 2F).

We next modelled the effect of non-specific binding on estimates OT-I TCR $K_D$ for active pMHC by extending the 1:1 binding model to include a second term to account for binding to inactive pMHC (Fig. 2G). This showed that, while having some inactive pMHCs (26%) would not distort $K_D$ estimates for OTI TCR binding to higher-affinity pMHC, inactive pMHC would appreciably affect $K_D$ estimates for lower-affinity pMHCs (Fig. 2H). As expected, a larger distortion would be observed with a higher fraction of inactive pMHC (Appendix Fig. S2A–C).

Using the apparent $K_D$ and the fraction of active pMHC, we were able to estimate the $K_D$ of OT-I TCR binding to active pMHC (Fig. 2I). To validate this new method, we used it to measure the affinity of OT-I binding to VSV pMHC and two different levels of partially denatured VSV pMHC (Fig. EV3A,B). The apparent $K_D$, estimated using the original method, displayed wide variation across these three surfaces, while on the other hand, our new method produced the same active $K_D$ value on all surfaces despite variations in the amount of denatured pMHC. This confirmed that our method can reliably estimate active $K_D$ values. When we applied this extended workflow to estimate the active $K_D$ for all pMHC, we found that it produced similar values for higher-affinity interactions but up to fourfold higher $K_D$ values for the lower-affinity interactions (Fig. 2J; Table 1).

## Revised OT-I affinities display much larger variation and weaker binding to self-pMHC

The revised affinities that we now report for OT-I TCR binding various pMHCs at 37 °C are very different from the values previously reported, and the discrepancies are most pronounced for low-affinity peptides (Fig. 3A). For example, the OT-I TCR was originally reported to bind E1 pMHC with a $K_D$ of 26.8 µM, whereas we now report a 160-fold lower affinity of 3637 µM. Our measurements are in closer agreement with five affinities more recently measured at 25 °C (Stepanek et al, 2014) (Fig. 3B). Importantly, our revised OT-I/pMHC affinities show much greater variation than previously reported. For example, we find a 150-fold variation in $K_D$ from 34 µM to 5066 µM for the N4 and R4 pMHC, respectively. In contrast, Alam et al (Alam et al, 1999) described only a twofold difference between OT-I TCR affinities for N4 and R4 pMHCs.

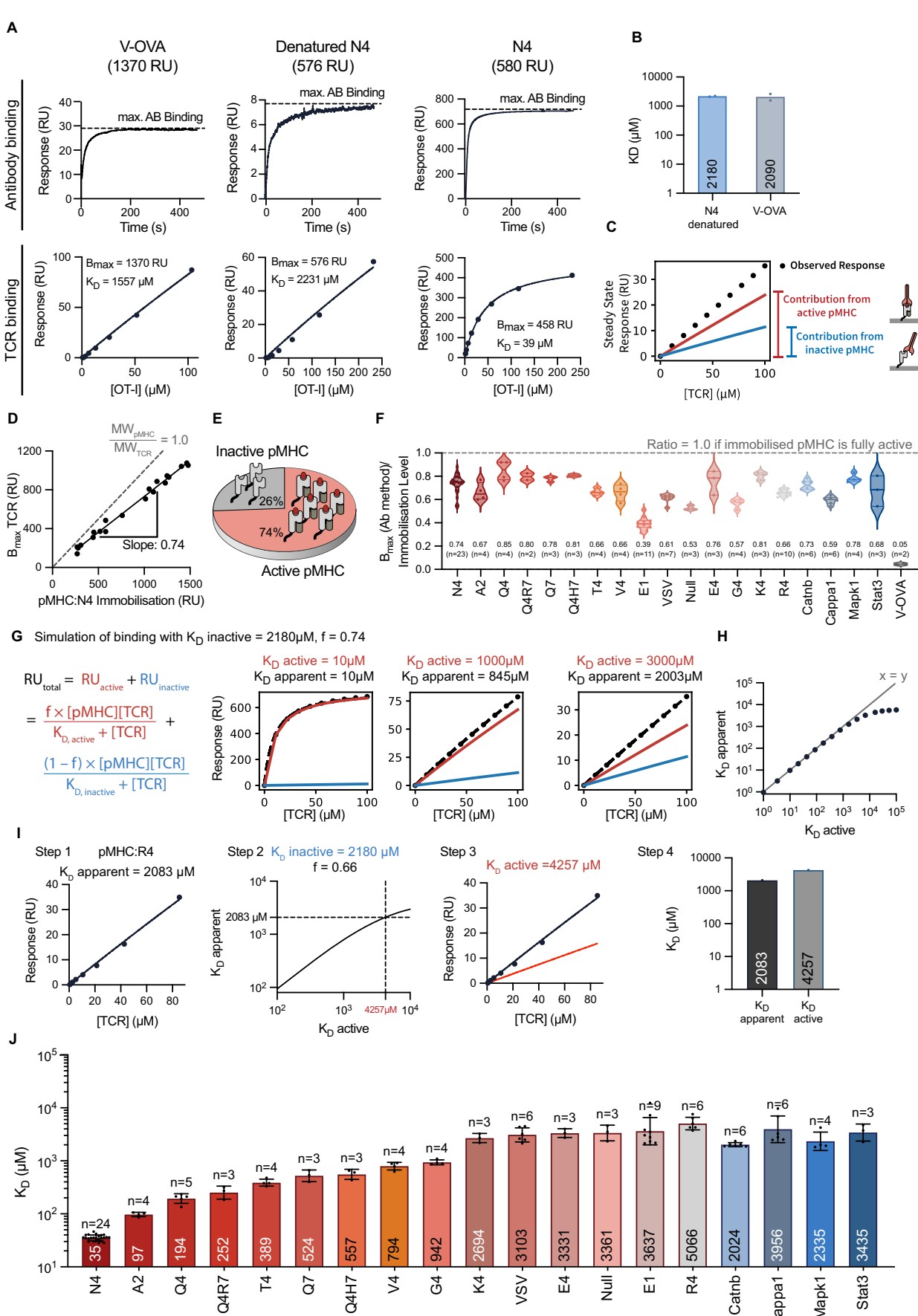

◄ **Figure 2. The presence of inactive pMHC can impact TCR binding and $K_D$ estimates for low-affinity interactions.**

(A) SPR sensorgram of B2M antibody binding (top) and OT-I TCR steady-state binding curve (bottom). The steady-state binding for denatured N4 and V-OVA was fit with a 1:1 model to estimate $K_D$ with $B_{max}$ constrained to the pMHC immobilisation level. A low pH glycine solution denatured N4. (B) Estimated $K_D$ for the data in (A) for $N = 2$ independent SPR experiments. (C) Schematic of the overall observed binding decomposed into the contribution from active and inactive pMHC. (D) The OT-I $B_{max}$ over the immobilisation level of N4 produces a slope of 0.74 ($N = 20$). A slope of 1.0 is expected for a 1:1 interaction if all pMHC (49 kDa) is active and can bind the TCR (51 kDa). (E) Schematic showing that 74% of N4 pMHC is active and can bind the TCR. (F) Ratio of $B_{max}$ (obtained from the standard curve) to pMHC immobilisation determines the fraction of active pMHC for all peptides tested. Mean and sample size ($n$) are given below data points. (G) Simulated TCR binding to surfaces with inactive and active pMHC using a fixed fraction of active pMHCs with different affinities (columns). The overall binding (black) is fit to a 1:1 model to estimate the apparent $K_D$. (H) The fitted apparent $K_D$ over the true active $K_D$. (I) Workflow applied to convert the apparent $K_D$ of OT-I/R4 interaction (2083 µM) into the active $K_D$ (4257 µM). (J) Revised OT-I active $K_D$ values for the indicated peptide at 37 °C (geometric mean $K_D$ indicated within bar, also see Table 1 for geometric mean $K_D$ +/− geometric SD.) $N = X$ SPR experiments were performed. Source data are available online for this figure.

Although self-peptides have been identified for the OT-I TCR, previous estimates of their affinities have only been performed at the unphysiologically low temperature of 10 °C (Juang et al, 2010). Using our method, we show that the OT-I TCR binds Catnb and Cappa1 self-peptides with $K_D$ values of 2024 µM and 3956 µM, respectively, at 37 °C, which are appreciably larger than the $K_D$ values of 136 µM and 211 µM reported at 10 °C (Fig. 3C). Whereas the $K_D$ varied by 24-fold between the foreign (N4) and self (Cappa1) antigens at 10 °C, we now find a much larger variation of 115-fold at 37 °C. This suggests that thymic positive selection can proceed with ultra-low affinities, enabling a much larger affinity window between foreign and self-antigens in the periphery.

## Revised 3D affinities correlate well with 2D affinities

The process of antigen recognition takes place at the T cell–APC contact interface, where both TCRs and pMHCs are attached to membranes that confine their movements to two dimensions. It has long been speculated that the 2D and 3D TCR/pMHC binding properties may not correlate well because 2D receptor/ligand interactions at cellular interfaces are subjected to a number of processes, including spatial redistribution and/or forces, that are not present in 3D solution measurements (Van der Merwe, 2001). In apparent support of this, original 3D affinity values for the OT-I TCR displayed a highly non-linear power relationship with a power (or slope on log-transformed values) of 2.9 (Fig. 4A). In other words, small changes in 3D affinity were associated with large changes in 2D affinity. This suggested that pulling forces on the TCR/pMHC may improve antigen discrimination (Huang et al, 2010; Liu et al, 2014). In contrast, our revised 3D affinity values correlate well with 2D affinity values, with a much shallower slope of 1.3 (Fig. 4B,C). These new findings are similar to those reported for the 1E6 TCR, where the 3D and 2D affinities correlate very well, with a slope of 1.0 (i.e., linear correlation, Fig. EV4) (Cole et al, 2016). The linear correlation between 2D and 3D affinity for the OT-I and 1E6 TCRs argues against a substantial effect of interface processes, such as forces, on the 2D TCR/pMHC affinities in these settings.

## The OT-I TCR displays enhanced but imperfect antigen discrimination

Given the large differences between the originally reported OT-I affinities and the revised ones reported here, we measured the potency of these peptides in functional assays. Using naive $CD8^+$ T cells from OT-I TCR transgenic mice, we quantified T-cell activation potency in terms of CD69

upregulation for eight different peptides (Fig. 5A). There was a significant correlation between peptide potency ($EC_{50}$) and our revised $K_D$ values but not with the original $K_D$ values (Fig. 5B). Another key difference was the discriminatory power, which is obtained by the slope of the relationship. The original $K_D$ measurements produced a steep slope ($\alpha = 16$). This indicates that a small reduction in OT-I/pMHC affinity would abolish the T cell response, which has been termed near-perfect or absolute discrimination (Altan-Bonnet and Germain, 2005; Francois et al, 2013; Ganti et al, 2020). In contrast, our revised $K_D$ measurements produced a more modest discriminatory power ($\alpha = 2.4$). We previously defined categories of discrimination based on the value of $\alpha$: baseline discrimination ($\alpha \sim 1.0$), enhanced discrimination ($\alpha > 1.0$), and near-perfect discrimination ($\alpha > 9.0$) (Pettmann et al, 2021). Using this terminology, we conclude that the OT-I TCR displays enhanced but imperfect antigen discrimination.

We next plotted the potency data from seven previous functional studies against our revised $K_D$ measurements, which measured T-cell activation via target cell lysis (Hogquist et al, 1995), surface receptor expression (Daniels et al, 2006; Lo et al, 2023, 2019; Rosette et al, 2001), T-cell proliferation (Huang et al, 2010) and cytokine production (Zehn et al, 2009) (Fig. 5C–I). We found that, whereas the original $K_D$ measurements correlated poorly with potency, and indicated a near-perfect discriminatory power of ∼20.3, our revised $K_D$ measurements correlated well with potency and indicated enhanced but imperfect discriminatory power of ∼2.4 (Fig. 5J). This conclusion holds when discriminatory power is calculated using apparent $K_D$ values, confirming that our method to determine active $K_D$ values does not affect our conclusion (Fig. EV5A–I).

The self-peptides Catnb and Cappa1 have recently been shown to activate OT-I T cells at very high concentrations (∼100 µM) (Lo et al, 2023). This is consistent with their very low affinities (Fig. 3J). Indeed, like Catnb and Cappa1, other very low-affinity peptides such as E1 and R4 have also been shown to induce positive selection (Hogquist et al, 1995, 1992). This suggests that, while T cells can maintain tolerance to self-pMHCs ($K_D > 2000$ µM) when expressed at normal levels, this tolerance can be broken if these self-peptides are abnormally overexpressed.

## The discriminatory power of the OT-I TCR increases with $K_D$, consistent with the kinetic proofreading mechanism

The discriminatory power is an empirical measure of antigen discrimination, and a mechanistic description is provided by the kinetic proofreading model (Fig. 6A). In this model, a time delay ($\tau_{kp}$) between pMHC binding and TCR signalling produced by a series of

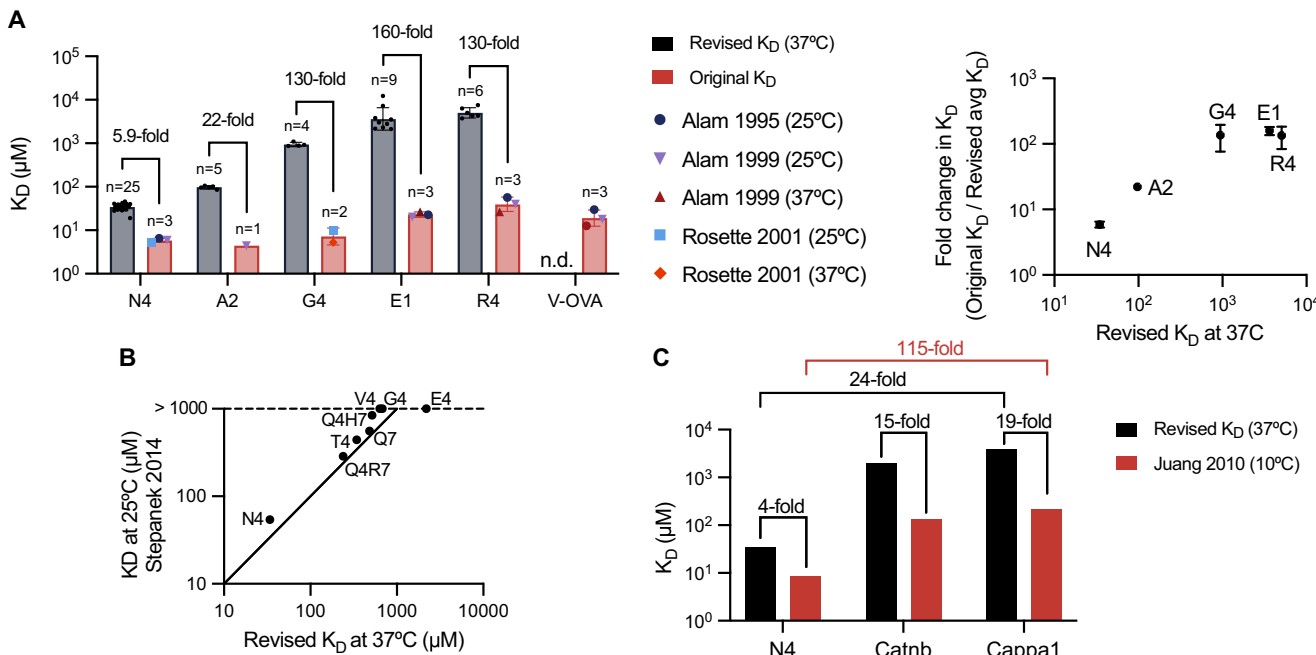

**Figure 3.  Discrepancies between published and revised OT-I affinities.**

(**A**) Comparison between original OT-I $K_D$ values measured by SPR at 25 °C and 37 °C (Alam et al, 1999, 1996; Rosette et al, 2001) and the revised $K_D$ values measured at 37 °C in the present work as bar graphs (left) or scatter plot of fold change (right). A description of how original $K_D$ values were obtained from previous publications can be found in the method section. The original measurements at 37 °C for N4 and A2 were excluded because they displayed biphasic binding, making affinity and kinetic estimates unreliable. Graphs show geometric mean $K_D$ values $+/-$ geometric SD. (**B**) Comparison of $K_D$ values determined at 25 °C (Stepanek et al, 2014) with our revised $K_D$ values at 37 °C. Solid line indicates the identity line (y = x). (**C**) Comparison of $K_D$ values measured for self-peptides at 10 °C (Juang et al, 2010) and our revised $K_D$ values at 37 °C. Source data are available online for this figure.

biochemical signalling steps ($N$, each with rate $k_P$) reduces the probability of productive signalling by pMHCs with a short dwell time. To estimate these parameters, we fit the model to the averaged potency of each OT-I ligand using our revised 3D affinity measurements (Fig. 6B). Although the number of proofreading steps for the OT-I TCR ($N = 4.2$) is larger than previous reports for the 1G4 TCR and a CAR ($N < 3$) (Pettmann et al, 2021; Tischer and Weiner, 2019), the overall time delay is shorter ($\tau_{kp} = 0.21$ s for OT-I vs 2.7 s for 1G4) because of a much higher rate of traversing each step ($21$ s$^{-1}$ for OT-I vs $1.0$ s$^{-1}$ for the 1G4). These fitted proofreading parameters that explain a discriminatory power of 2.4 are also consistent with the high antigen sensitivity of the TCR (Fig. 6C).

We noted that when plotting all the potency data against affinity, a non-linear relationship emerges (Fig. 6B). This indicates that the discriminatory power law relationship between potency and affinity has a discriminatory power that varies with affinity. Indeed, a plot of discriminatory power against affinity shows that it increases from near 0 at low $K_D$ to 4 at high $K_D$ values (Fig. 6D). In other words, the discriminatory power of the OT-I TCR is highest for lower-affinity antigens, a property predicted by the kinetic proofreading model.

## Discussion

Despite the fact that OT-I TCR transgenic mice are widely used, there was no accurate affinity data for the OT-I TCR binding

relevant pMHCs at physiological temperatures, and original measurements showed unusual biphasic binding at 37 °C. These original measurements gave rise to the notion that the OT-I TCR displays near-perfect antigen discrimination based on affinity and that the 3D affinity measurements do not correlate with 2D affinity measurements. Here, we developed a new method to accurately measure ultra-low affinities and used it to systematically measure the OT-I TCR affinity to 19 commonly used peptides. We found that our revised 3D $K_D$ values correlate well with 2D $K_D$ values, and with T-cell functional responses. Importantly, these $K_D$ values, together with functional data from many laboratories, demonstrate that the OT-I TCR displays enhanced but imperfect antigen discrimination, as has been reported for other TCRs (Pettmann et al, 2021). Finally, we have shown that the discriminatory power of the OT-I TCR is highest for low-affinity pMHC ligands, a result explained by the kinetic proofreading model.

In control experiments, we found that the OT-I TCR binds non-specifically to unfolded MHC, and we estimated this affinity to be $K_D \sim 2000 \mu M$ (Fig. 2). This binding may represent binding to the empty, relatively non-specific peptide-binding groove in unfolded pMHC. Consistent with this, it has been observed that a TCR can weakly bind empty MHC molecules but not those loaded with irrelevant peptides (Moritz et al, 2019). When the fraction of inactive pMHC is small, their presence is unlikely to impact the accuracy of higher-affinity measurements. However, even a small fraction of inactive pMHC can be problematic when measuring very low affinities. We developed a simple workflow to quantify the

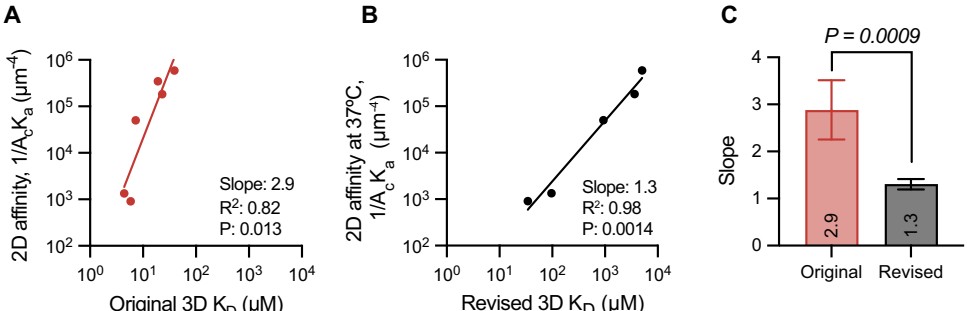

**Figure 4.  Revised 3D affinities produce high correlations with 2D affinities and display similar variation.**

(A, B) Correlation between 2D affinity values (Huang et al, 2010) with the (A) original ($N = 6$) and (B) our revised 3D $K_D$ values ($N = 5$). (C) Fitted slopes $+/-$ standard errors from linear regression fit in (A, B). An $F$ test determines the $P$ value for the null hypothesis that a single slope can fit both correlations. Source data are available online for this figure.

amount of inactive pMHC enabling us to extract the $K_D$ for active pMHC from the apparent $K_D$ values. This conversion relied on an accurate estimate of the amount of inactive pMHC, which we estimated by injection of conformationally sensitive antibodies to pMHC at the end of each experiment. We demonstrated that we can extract the same active $K_D$ values from OT-I binding to surfaces with different levels of inactive pMHC (Fig. EV3A,B). Interestingly, a loss of peptide and $\beta_2$m from cell surface MHC class I has been suggested to induce homodimerisation via the $\alpha 3$ domains, followed by internalisation (Dirscherl et al, 2022). This would reduce the likelihood that unfolded MHC on the APC would bind to TCRs or other receptors, activating cells non-specifically.

We found a linear correlation between the TCR/pMHC affinities measured using purified proteins in solution (i.e., 3D $K_D$) and previously reported 2D affinities (Fig. 4). This contrasts with the highly non-linear correlation between the 2D and the original reported 3D $K_D$ values (Huang et al, 2010). A linear correlation between 3D and 2D affinities has also been reported for the 1E6 TCR (Cole et al, 2016) (Fig. EV5A–I). These results suggest that, despite differences between the TCR/pMHC interaction in solution and within cell-cell interfaces, T cells accurately measure linear proxies of the 3D $K_D$. This is unexpected because T cells generate large mechanical forces during antigen recognition (Colin-York et al, 2019; Husson et al, 2011) and molecular forces can have large non-linear effects on the TCR/pMHC bond lifetime (Allard et al, 2012; Chen and Zhu, 2013; Klotzsch and Schütz, 2013; Liu et al, 2014; Rogers et al, 2024). These results can be reconciled by the force-shielding model (Pettmann et al, 2023), which proposes that T cells deploy mechanisms to shield the TCR/pMHC interaction from molecular forces. By eliminating force, the 2D TCR/pMHC lifetimes would be expected to correlate with the 3D lifetimes or 3D $K_D$ (when $k_{on}$ displays minimal variation). Consistent with this model, the 2D and 3D lifetimes have been shown to be similar (O'Donoghue et al, 2013), and most TCR/pMHC interactions appear to take place without experiencing forces (Schrangl et al, 2025). We have suggested that receptor/ligand interactions, such as CD2/CD58 and/or LFA-1/ICAM-1, mediate force-shielding and this would require spatial redistribution near the TCR/pMHC (Pettmann et al, 2023). This is supported by the observation that ligand mobility is required to abolish TCR/pMHC forces (Göhring et al, 2021). Taken together, the high linear correlation between 3D

and 2D $K_D$ values suggests that, although T cells generate large mechanical forces during the process of antigen recognition, they deploy mechanisms to shield TCR/pMHC interactions from these forces. A limitation of this conclusion is that we cannot rule out the possibility that a non-linear relationship between the 3D and 2D on-rate and off-rates cancel each other out to generate an apparent linear relationship between the 3D and 2D affinities.

The original $K_D$ values reported for OT-I suggested that T cells possessed near-perfect discriminatory powers of $\sim$5–40 (Fig. 5J). For example, while the OT-I TCR bound the E1 peptide with a threefold lower affinity than the N4 peptide, OT-I T cells required a 100,000-fold higher concentration of E1 than N4 peptides to be activated. In contrast, other TCRs exhibit a discriminatory power of approximately 2, where T cells require only $3^2 = 9$-fold higher concentrations to be activated by peptide antigens with a 3-fold lower affinity (Pettmann et al, 2021). Our revised affinity measurements show that the OT-I TCR binds the E1 peptide with a 106-fold lower affinity than the N4 peptide (Fig. 2J), rather than the previously reported threefold lower affinity. Thus, OT-I T cells failed to respond to E1, not because of their exceptionally high discriminatory power, but because the OT-I TCR binds E1 with exceptionally low affinity. Using these $K_D$ values we calculate the discriminatory power of OT-I TCR to be $\sim$2.4, similar to other TCRs. We find that the discrimination power is consistent between different T-cell activation readouts, consistent with previous results (Pettmann et al, 2021). Our new results therefore resolve this major discrepancy between the OT-I TCR and other TCRs.

Like other murine TCRs (discriminatory power 3.2), the OT-I TCR appears to have a greater discriminatory power than human TCRs (2.4 vs 2.0) (Pettmann et al, 2021). It also has different kinetic proofreading parameters with a time delay that is more than tenfold shorter than the human 1G4 TCR (2.7 s compared to 0.21 s for OT-I) (Pettmann et al, 2021). A key difference between studies on murine and human TCRs is that the murine T cells, like the OT-I, are often obtained from TCR transgenic mice where they have undergone development while expressing the TCR. In contrast, human TCRs are often studied after transfection into polyclonal primary T cells. Since T cells undergo tuning during development based on the TCRs they express (Cho and Sprent, 2018), this is likely to affect their sensitivity and discriminatory power. Although incompletely understood, one mechanism for tuning is altering

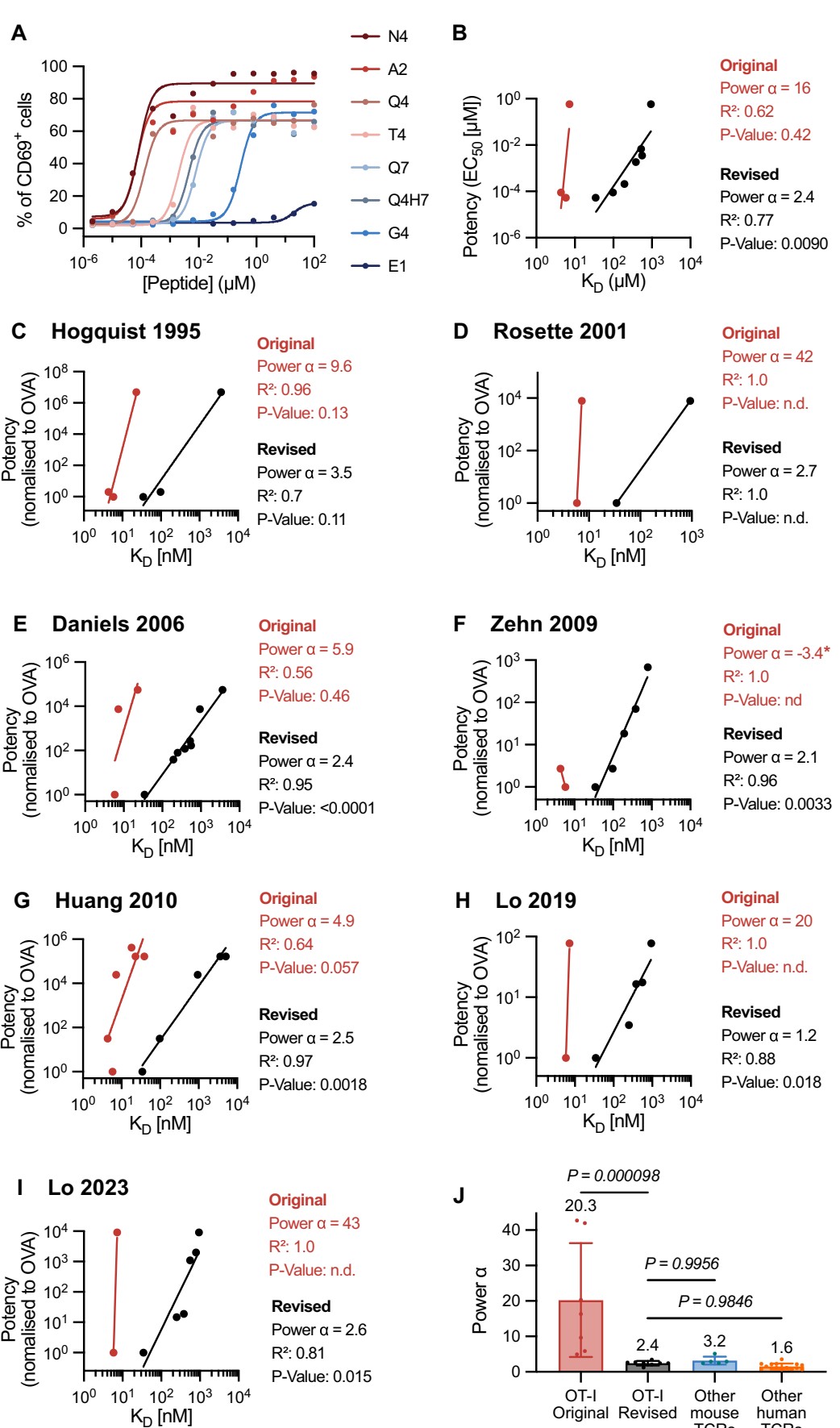

**Figure 5. Revised 3D affinities correlate with OT-I T-cell responses and reveal enhanced but imperfect antigen discrimination.**

(A, B) Representative OT-I T cell activation by the indicated peptides (A) and peptide potency ($EC_{50}$) over $K_D$ (B) for $N = 2$ biological repeats. (C–I) Published potency data from the indicated study over original or revised $K_D$ values. A detailed description of how the potency data was obtained can be found in "Methods". A power law (potency ~ $(K_D)^a$) is fit to the data to estimate the discriminatory power ($a$). A Pearson correlation is used to determine $R^2$ and $P$ values on log-transformed values. The original $K_D$ values are the average $K_D$ values from Fig. 3A, and the number of data points between the original and revised plots can differ because some peptides do not have an original $K_D$ measurement. (J) The discriminatory power from panels B-I (OT-I original and revised) and from other mouse and human TCRs (see Pettmann et al (Pettmann et al, 2021)). Negative values for $a$ as observed for (F) were excluded. One-way ANOVA was used to determine $P$ values. Sample sizes: OT-I original $N = 7$, OT-I revised $N = 8$, other mouse TCRs $N = 5$, other human TCRs $N = 15$. Bars show mean values (also indicated above the bar) ± SD. Source data are available online for this figure.

expression surface CD5 levels (Persaud et al, 2014), which we have shown modifies TCR discrimination (Cabezas-Caballero et al, 2025). The fact that the OT-I has the shortest 3D half-life ever reported for a TCR (Pettmann et al, 2023), and the resulting tuning of signalling by OT-I TCR expressing T cells to adapt to this short half-life, may contribute to the difference between its kinetic proofreading parameters and those observed for the 1G4 TCR expressed in polyclonal human T cells.

The enhanced but imperfect discriminatory power of the OT-I TCR and the ultra-low affinities that we now report for self-antigens have two important implications for peripheral tolerance. First, it suggests that ultra-low-affinity self-antigens can induce thymic positive selection and as a result, the ~100-fold affinity window between foreign and self-antigens is likely much larger than previously believed (Fig. 3C). Second, it suggests that peripheral tolerance can be broken by self-pMHCs provided they are expressed at sufficiently high concentrations. Indeed, the self-peptides Catnb and Cappa1 can activate OT-I T cells when presented at ~$10^5$-fold higher concentrations compared to the foreign antigen N4 (10 vs $10^{-4}$ μM) (Lo et al, 2023). This is consistent with an imperfect discriminatory power of 2.4 that would predict T-cell responses by these self-peptides when their concentration increases by ~$100^{2.4} = 63,000$. This supports the notion that, in addition to recognising foreign antigens, T cells have a homoeostatic surveillance role in detecting cells expressing aberrantly high levels of protein, such as hormone-secreting endocrine tumours (Korem Kohanim et al, 2020).

One prediction of the kinetic-proofreading model is that the discriminatory power of a TCR should increase with $K_D$, i.e., discrimination is largest for lower-affinity pMHCs (Pettmann et al, 2021). This prediction has been difficult to test in the absence of sufficient TCR/pMHCs measurements over a sufficiently wide range of $K_D$ values. The large number of OT-I functional studies using pMHCs whose affinities we now report to have a wide range of $K_D$ values has enabled us to accurately measure how the discriminatory power varies with $K_D$ for the first time (Fig. 6D). This has highlighted that the discriminatory power can be as high as ~4 at the ultra-low-affinity range whereas it decreases to values below 1 at the higher-affinity range where potency begins to saturate. Additional studies are required to establish whether this is a general feature of TCR discrimination.

We have confirmed that kinetic proofreading, i.e., a time delay between antigen binding and productive TCR signalling, is necessary to explain antigen discrimination by OT-I T cells. Our findings are broadly consistent with other studies that have applied the kinetic proofreading model to T-cell activation data,

encompassing both downstream functional responses (Pettmann et al, 2021) and more proximal signalling events such as calcium flux (Yousefi et al, 2019) and PLC-γ activity (Tischer and Weiner, 2019). While these studies collectively support the relevance of kinetic proofreading in T cell activation, future work is needed to identify the precise molecular steps that mediate this process. Current evidence points to roles for TCR-CD3 phosphorylation, ZAP-70 recruitment, and LAT phosphorylation in enforcing the time delay required for ligand discrimination (Courtney et al, 2019; Goyette et al, 2022; Lo et al, 2019; McAffee et al, 2022). In addition, we have shown that co-signalling receptors, including CD8, CD4, and CD5, while not formal proofreading steps, can nonetheless influence ligand discrimination in primary human CD8+ T cells, suggesting that they control the rate of one or more steps in the process (Cabezas-Caballero et al, 2025).

The method we have introduced to accurately measure ultra-low-affinity TCR/pMHC interactions has several important future applications. First, although our focus in this study and in previous work (Pettmann et al, 2021) has been on CD8+ T cells, it will be crucial to extend this approach to CD4+ T cells, such as OT-II T cells, to determine their discriminatory power with comparable precision. Second, a general understanding of the affinity differences between self and non-self-peptides remains elusive. While numerous autoreactive human and mouse TCRs have been identified, a major limitation has been the ability to measure the very low affinities they exhibit with self-pMHCs (Yin et al, 2012). Accurate affinity measurements are still lacking in many cases, despite clear functional responses observed at high peptide concentrations. Lastly, most known TCR interactions with self-pMHCs have been identified in the context of autoimmune diseases, making it unclear whether these represent typical or exceptional cases. Moving forward, our method offers the potential to precisely measure affinities between wild-type TCRs and their candidate self-pMHCs, shedding light on the broader landscape of T cell self vs non-self-discrimination.

In conclusion, by accurately measuring 3D $K_D$ values between the OT-I and 19 pMHCs, including foreign and self-antigens, we have reconciled reported discrepancies between the OT-I TCR and other TCRs, confirmed that the OT-I displays physiological affinity to its foreign antigen, shown that the 3D affinity predicts the 2D affinity, and that the OT-I TCR displays enhanced but imperfect discrimination, which increases for lower-affinity antigens. Collectively, these results highlight that despite the complex and mechanically active T cell/APC interface, T cells make decisions based on proxies for the 3D $K_D$ measured with purified proteins in solution.

# Methods

### Reagents and tools table

| Reagent/resource | Reference or source | Identifier or catalogue number |
|---|---|---|
| **Experimental models** | | |
| OT-I mice | Jackson Laboratory | 003831 |
| CD45.1 mice | Charles River | 708 |
| **Recombinant DNA** | | |
| pGMT7-OTIa-HC (OT-I alpha chain) | Stepanek et al (2014) | |
| pGMT7-OTIb-HC (OT-I beta chain) | This study | |
| pGMT7-OTIb-HC-His | This study | |
| pET14b-H2Kb | Vincenzo Cerundolo | |
| pET14b-H2Kb-AviTag | This study | |
| pTO-N-human-beta2m | Ricardo Fernandes? https://www.pnas.org/doi/epdf/10.1073/pnas.89.8.3429? | |
| **Antibodies** | | |
| Anti-Mouse H-2Kb (Clone: Y-3) | Leinco Technologies | RRID: AB_2737575 |
| Anti-human beta-2 Microglobulin (Clone: B2M-01) | Thermo Fisher Scientific | RRID: AB_1070702 |
| Anti-mouse CD45.1 (Clone: A20) FITC | Biolegend | 110705 |
| Anti-mouse CD8a (Clone: 53-6.7) PE | Biolegend | 100707 |
| Anti-mouse CD69 (Clone: H1.2F3) BV421 | Biolegend | 104527 |
| Anti-mouse CD44 (Clone: IM7) BV785 | Biolegend | 103041 |
| **Oligonucleotides and other sequence-based reagents** | | |
| Primer for adding His-Tag to OTIb-HC | FW: CACCACCACCA CCACCACTAATAAGAA TTCCGATCCGGCTGC; RV: GTCGGCGCGGCCCCATG | |
| Primer for adding AviTag to H2KB | FW: CTTCAAAAATATCGTT CAGGCCACGATGATTCC ACACCATTTTCTG; RV: CGCAGAAAATTGAATG GCATGAATAAAAGCTTGCGGCCGC | |
| **Chemicals, enzymes and other reagents** | | |
| Amine coupling kit | Cytiva | BR100050 |
| Ampicillin | Cayman | 14417 |
| BirA biotin-protein ligase bulk reaction kit | Avidity | |
| BugBuster Protein Extraction Reagent | Millipore | 70584-M |
| Series S Sensor Chip CM5 | Cytiva | |
| DMSO | Sigma | D2650 |
| HBS-EP | Cytiva | BR100669 |

| Reagent/resource | Reference or source | Identifier or catalogue number |
|---|---|---|
| IPTG | Thermo Scientific | R0391 |
| LB | | |
| QIAprep Spin Miniprep Kit | Qiagen | 27104 |
| PBS | Sigma-Aldrich | D8537 |
| Q5 Site-Directed Mutagenesis Kit | NEB | E0554S |
| Trizma base | Sigma | T6066 |
| Urea | VWR | 28877.292 |
| Arg-HCl | Sigma | A5131 |
| L-Glutathione oxidised | Cayman | 35825 |
| L-Glutathione reduced | Fisher Scientific | 15494589 |
| MojoSortTM CD8 + T-cell negative isolation kit and magnets | Biolegend | 480008 and 480019 |
| RPMI 1640 | Gibco | 21870-076 |
| FCS | Sigma | F9665 |
| Penicillin-Streptomycin | Gibco | 10378-016 |
| Zombie NIR Fixable Viability Kit | Biolegend | 423106/423105 |
| TruStain FcXTM (anti-mouse CD16/32) | Biolegend | 101319 |
| HiTrap Q HP column | Cytiva | 17115401 |
| Superdex 200 Increase 10/300 GL column | Cytiva | 28990944 |
| Superdex 75 10/300 GL column | GE Healthcare* | 17-5174-01 |
| *E. coli* DH5α | In-house production | |
| *E. coli* BL21(DE3) | In-house production | |
| **Software** | | |
| GraphPad Prism (v10) | GraphPad Software | |
| SnapGene (v4) | GSL Biotech LLC | |
| BiaEvaluation (v4.1) | GE Healthcare* | |
| Python (3.7.4) | | |
| BDFACSDiva (v8.0) | | |
| FlowJoTM software (v10.4.2) | Tree Star | |
| **Other** | | |
| Soluble biotinylated Peptide-MHC | NIH Tetramer Facility | |
| Synthetic peptides | Peptide Protein Research or GenScript | Custom synthesis |
| ÄKTA Pure™ chromatography system | GE Healthcare* | |
| BiaCore T200 SPR System | GE Healthcare* | |

| Reagent/resource | Reference or source | Identifier or catalogue number |
|---|---|---|
| BD LSR II Flow Cytometer | BD Bioscience | |
| FortessaX20 Flow Cytometer | BD Bioscience | |

*Currently Cytiva.

The amino acid sequences of proteins used in this work can be found in Table 2.

## Protein expression and purification

### OT-I TCR

For affinity measurements with soluble TCR, we used an OT-I TCR construct consisting of the murine variable OT-I domain and the human constant domain truncated above the transmembrane domain with an artificial interchain disulphide, as described previously (Stepanek et al, 2014). TCR $\alpha$ and $\beta$ chains were expressed in BL21 DE3 *Escherichia coli* cells following induction with 0.15 mM IPTG and isolated from inclusion bodies. Proteins were stored at −80 °C until use.

OT-I-TCR was refolded by adding 15 mg of each chain dropwise in 1 L refolding buffer (150 mM Tris-HCl (pH 8.0), 3 M Urea, 200 mM Arg-HCl, 0.5 mM EDTA, 0.1 mM PMSF), followed by dialysis for 3 days in 10 L Tris buffer (10 mM Tris-HCl (pH 8.5)), with a buffer change after 24 h. After dialysis, the protein was filtered and purified using ion-exchange chromatography (HiTrap Q column [Cytiva]) with a NaCl gradient in the dialysis buffer. Next, protein was concentrated and purified again by size-exclusion chromatography (Superdex 200 Increase column [Cytiva]) in HBS-EP buffer (0.01 M HEPES pH 7.4, 0.15 M NaCl, 3 mM EDTA, 0.005% v/v Tween-20). Purified TCR was used for SPR measurements not longer than 24 h after purification to avoid aggregation. Protein concentration was measured with Nanodrop.

### pMHCs

Class I pMHCs were generated using mouse H-2K$^b$ heavy chain and human beta-2 microglobulin ($\beta$2 m), biotinylated on the C terminus of the heavy chain. pMHCs produced in HEK293T cells were biotinylated and peptide-exchanged by the NIH tetramer facility. For pMHC produced in *E.coli*, soluble mouse H2K$^b$ heavy chain with a C-terminal AviTag/BirA recognition sequence and human $\beta$-2m were expressed separately in BL21 DE3 *E. coli* cells and isolated from inclusion bodies. MHC heavy chain, $\beta$-2m and peptide were then added dropwise to the refolding buffer (100 mM Tris-HCl, pH 8.0, 400 mM L-Arg·HCl, 2 mM EDTA, 5 mM reduced glutathione, 0.5 mM Oxidised glutathione, 0.1 mM PMSF) at a concentration of 2 μM, 1 μM, 10 μM, respectively. The protein solution was kept under constant stirring for 48 h at 4 °C. Afterwards, the refold was filtered through a 0.45 μL filter and concentrated using centrifugal filters. pMHCs were biotinylated overnight at room temperature using the BirA Biotin-protein ligase bulk reaction kit (Avidity LLC). Next, pMHCs were purified by size-exclusion chromatography (Superdex 75 column [GE Healthcare]) in HBS-EP Buffer. pMHCs were aliquoted and stored at −80 °C until use.

## Surface plasmon resonance

Affinities of the OT-I TCR to peptide variants were measured with a newly established SPR technique (SPR) for ultra-low TCR-pMHC affinities described previously (Pettmann et al, 2021). We generated soluble OTI TCR, which recognises the ovalbumin (OVA) peptide (SIINFEKL) loaded onto a murine H-2K$^b$ class I MHC. Equilibrium binding analysis of TCR-pMHC interactions was performed by SPR on a Biacore T200 instrument (GE Healthcare Life Sciences) with CM5 sensor chips. HBS-EP was used as a running buffer, and all $K_D$ measurements were performed at 37 °C. For protein immobilisation, the sensor chip was saturated with streptavidin using an amino coupling kit (Cytiva). Biotinylated pMHCs were injected into experimental flow cells (FC) for different durations to immobilise 400–1500 RU pMHC. Matching levels of CD86 were immobilised in FC1 as a reference. Next, excess streptavidin was blocked with two 40 s injections of 500 μM biotin (Avidity) and the sensor was conditioned with 8 injections of running buffer. TCR was injected at increasing concentrations at 30 μl/min. Buffer was injected after every 2 or 3 TCR injections. Following TCR injections, anti-$\beta$2m antibody (B2M-01 Invitrogen, MA1-19141) that binds correctly folded pMHC was injected for 8 min at 10 μl/min.

### Obtaining $K_D$ (apparent) from SPR data

Apparent $K_D$ values were obtained by fitting a 1:1 binding model (RUeq = $B_{max}$ ·[TCR]/($K_D$ + [TCR])) to the double referenced equilibrium RU values. For low-affinity antigens, this curve does not saturate at the highest TCR concentration, therefore an accurate prediction of $B_{max}$ and thus $K_D$ is not possible. Instead, the high-affinity N4 pMHC-TCR interaction was used to generate the empirical standard curve to relate the $B_{max}$ of TCR binding to the maximal antibody binding. For low-affinity peptides, $B_{max}$ was constrained to $B_{max}$ inferred from the standard curve when fitting the SPR data.

### Simulation of TCR Binding to mixed populations of active and inactive pMHC

Steady-state binding response of the TCR interacting with a mixture of active (correctly folded) and inactive pMHC was modelled using the following equation:

$$RU_{total} = RU_{active} + RU_{inactive}$$
$$= \frac{[TCR] \times [pMHC] \times f}{K_{D(active)} + [TCR]} + \frac{[TCR] \times [pMHC] \times (1-f)}{K_{D(active)} + [TCR]} \quad (1)$$

Here, $f$ is the fraction of active pMHC, [TCR] and [pMHC] are the concentrations of TCR and pMHC, respectively, $K_{D(active)}$ is the dissociation constant for the active pMHC-TCR interaction, and $K_{D(inactive)}$ is the dissociation constant for the inactive pMHC-TCR interaction.

### Determination of the fraction of active pMHC

The fraction of active pMHC ($f$) was determined from SPR measurements by comparing the TCR $B_{max}$ to the pMHC immobilisation level. The TCR $B_{max}$ was obtained using the B2M-01 antibody and the empirical standard curve in Fig. 1B (as

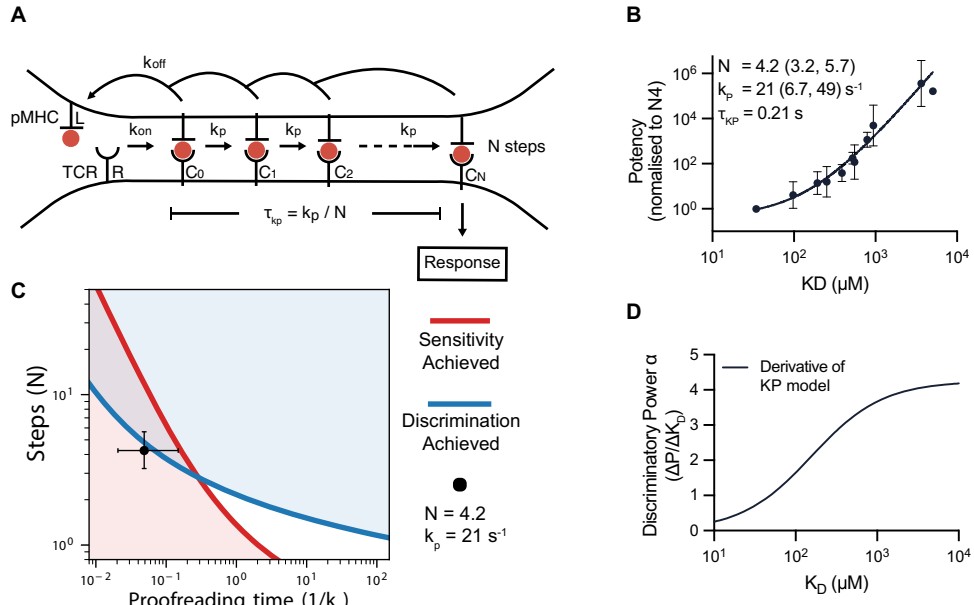

**Figure 6.  The kinetic proofreading model explains antigen discrimination by OT-I T cells with a short proofreading time delay and highlights that the discriminatory power can change with affinity.**

(A) Schematic of the kinetic proofreading model. (B) Potency over revised $K_D$ values fitted by the kinetic proofreading model (solid line). Potency data from all studies shown in Fig. 5 is normalised to N4 within each study before averaging across all studies and displayed as mean ± SD. The mean and 95%CI of the best-fit parameters are shown, $N = 40$ data points were included in the fit. (C) A binary heatmap displaying regions that achieve a discriminatory power of 2.4 (blue) and high antigen sensitivity (red). The fitted number of steps (N) and proofreading time (1/$k_p$) along with 95% CI is shown as a dot with error bars, which overlap with a region where both discrimination and sensitivity are achieved. The binary heatmap is produced as described in Pettmann et al (Pettmann et al, 2021) with the off-rate estimated as $k_{off} = K_D \cdot k_{on}$, where $K_D = 34$ μM and $k_{on} = 0.13$ μM$^{-1}$ s$^{-1}$ (for N4) and all other parameters as in Pettmann et al (Pettmann et al, 2021). (D) The discriminatory power (rate of change of potency with respect to $K_D$) over $K_D$ determined by taking the first derivative of the solid line in (B). Source data are available online for this figure.

described above). Thus,

$$f = \frac{\text{TCR } B_{\max}}{\text{Immobilisation Level}} \tag{2}$$

***Fitting SPR experimental data to obtain $K_{D(active)}$***

SPR experimental data were fitted with Eq.(1), with $K_{D(inactive)} = 2180$ μM and $f$ determined using Eq.(2). This procedure provided the $K_D$ value for the active fraction ($K_{D(active)}$).

***Interpolating $K_D$(active) from $K_D$(apparent)***

To relate $K_{D(active)}$ to $K_{D(apparent)}$, simulated binding curves were generated for a range of $K_{D(active)}$ values at a given active fraction $f$ using Eq.(1). These simulated datasets were then fitted with a 1:1 binding model ($RU_{eq} = B_{\max} \times [\text{TCR}]/(K_D + [\text{TCR}])$) to obtain $K_{D(apparent)}$. The resulting relationship between $K_{D(active)}$ and $K_{D(apparent)}$ was used to interpolate $K_{D(active)}$ for all peptides, given their experimentally determined $K_{D(apparent)}$ (Fig. EV1 and their average $f$ (Fig. 2F).

***Exclusion criteria***

If a given $K_{D(apparent)}$ fell outside the range of the $K_{D(active)}$-to-$K_{D(apparent)}$ curve (i.e., beyond the curve's saturation point), the data point was excluded. Under these circumstances, the observed TCR binding could be entirely explained by interactions with inactive pMHC alone. Applying this exclusion criterion removed one data point for peptide E1 from the final analysis.

## Mice

OT-I mice (JAX stock no.: 003831) were purchased from Jackson Laboratory, and CD45.1 mice from Charles River. Both male and female mice were used. Mice were bred and maintained in the University of Oxford specific pathogen-free (SPF) animal facilities. Mice were routinely screened for the absence of pathogens and were kept in individually ventilated cages with environmental enrichment at 20–24 °C, 45–65% humidity with a 12 h light/dark cycle (7am–7 pm) with half an hour dawn and dusk period. Mice were euthanized by CO2 asphyxiation followed by cervical dissociation. Breeding was conducted in agreement with the United Kingdom Animal Scientific Procedures Act of 1986 and performed under approved experimental procedures by the Home Office and the Local Ethics Reviews Committee (University of Oxford) under UK project licenses P4BEAEBB5 and PP3609558.

## T-cell activation

OT-I T cells were isolated from the lymph nodes and spleen of 6–12-week-old OT-I mice. Selection was carried out with a MojoSortTM CD8 + T-cell negative isolation kit and magnets (Biolegend, #480008 and #480019). Isolated OT-I T cells were resuspended in complete RPMI (RPMI 1640 [Gibco, #21870-076] supplemented with 2% FCS and 100x Penicillin-Streptomycin [Gibco, #10378-016]). Naive OT-I T cells (50,000) were seeded in 96-well U-bottom together with splenocytes (100,000) from CD45.1

**Table 2. Amino acid sequences for proteins used in this publication.**

| Name | Description | Sequence |
|---|---|---|
| OTI α | Soluble OTI α for expression in *E. coli*. Murine constant domain was replaced with human constant domain with additional cysteine (T159C) for improved refolding and stability. | MQQQVRQSPQ SLTVWEGETA ILNCSYEDST FNYFPWYQQF PGEGPALLIS IRSVSDKKED GRFTIFFNKR EKKLSLHITD SQPGDSATYF CAASDNYQLI WGSGTKLIIK PDIQNPDPAV YQLRDSKSSD KSVCLFTDFD SQTNVSQSKD SDVYITDKCV LDMRSMDFKS NSAVAWSNKS DFACANAFNN SIIPEDTFFP SPESS |
| OT-I β | Soluble OT-I β chain with 6x His-Tag on C terminus for expression in *E. coli*. Murine constant domain was replaced with human constant domain with additional cysteine (S169C) for improved refolding and stability. | MDSGVVQSPR HIIKEKGGRS VLTCIPISGH SNVVWYQQTL GKELKFLIQH YEKVERDKGF LPSRFSVQQF DDYHSEMNMS ALELEDSAMY FCASSRANYE QYFGPGTRLT VLEDLRNVFP PEVAVFEPSE AEISHTQKAT LVCLATGFYP DHVELSWWVN GKEVHSGVCT DPQPLKEQPA LNDSRYALSS RLRVSATFWQ DPRNHFRCQV QFYGLSENDE WTQDRAKPVT QIVSAEAWGR ADHHHHHH |
| H2Kᵇ heavy chain for *E. coli* | Soluble H2Kᵇ heavy chain with AviTag on C terminus for expression in *E. coli*. | MGPHSLRYFV TAVSRPGLGE PRYMEVGYVD DTEFVRFDSD AENPRYEPRA RWMEQEGPEY WERETQKAKG NEQSFRVDLR TLLGYYNQSK GGSHTIQVIS GCEVGSDGRL LRGYQQYAYD GCDYIALNED LKTWTAADMA ALITKHKWEQ AGEAERLRAY LEGTCVEWLR RYLKNGNATL LRTDSPKAHV THHSRPEDKV TLRCWALGFY PADITLTWQL NGEELIQDME LVETRPAGDG TFQKWASVVV PLGKEQYYTC HVYHQGLPEP LTLRWEPPPS GSLHHILDAQ KMVWNHRGLN DIFEAQKIEW HE |
| H2Kᵇ heavy chain for HEK293T cells | Soluble H2Kᵇ heavy chain with AviTag and His-Tag on C terminus for expression in HEK293T cells. Sequence from NIH Tetramer Facility. | GPHSLRYFVT AVSRPGLGEP RYMEVGYVDD TEFVRFDSDA ENPRYEPRAR WMEQEGPEYW ERETQKAKGN EQSFRVDLRT LLGYYNQSKG GSHTIQVISG CEVGSDGRLL RGYQQYAYDG CDYIALNEDL KTWTAADMAA LITKHKWEQA GEAERLRAYL EGTCVEWLRR YLKNGNATLL RTDSPKAHVT HHSRPEDKVT LRCWALGFYP ADITLTWQLN GEELIQDMEL VETRPAGDGT FQKWASVVVP LGKEQYYTCH VYHQGLPEPL TLRWEPPPST VSNMTSTTAP SAQLKKKLQA LKKKNAQLKW KLQALKKKLA QSGSGSGLND IFEAQKIEWH EHHHHHH |
| Human β2m | Human β2m for expression in *E. coli*. | MIQRTPKIQV YSRHPAENGK SNFLNCYVSG FHPSDIEVDL LKNGERIEKV EHSDLSFSKDW SFYLLYYTEF TPTEKDEYAC RVNHVTLSQP KIVKWDRDM |

mice loaded with the indicated dose of the following peptides: N4 (SIINFEKL), A2 (SAINFEKL), Q4 (SIIQFEKL), T4 (SIITFEKL), Q7 (SIINFEQL), Q4H7 (SIIQFEHL), G4 (SIIGFEKL), E1 (EIINFEKL). Cells were harvested after 24 h.

## Flow cytometry

Single-cell suspensions obtained from spleen or cultured CD8 + T cells were stained in V-bottom 96-well plates in flow cytometry buffer (2% FCS, 2 mM EDTA, and 0.02% sodium azide in 1× PBS). Live-dead staining and surface staining were performed using Zombie NIR Fixable Viability Kit (Biolegend, #423106/ 423105), TruStain FcXTM (anti-mouse CD16/32, Biolegend, #101319) and fluorochrome-conjugated primary antibodies against CD45.1 (Biolegend, clone: A20), CD8 (Biolegend, clone: 53-6.7), CD69 (Biolegend, clone: H1.2F3), and CD44 (Biolegend, clone: IM7). Cells were fixed using 4% PFA for 30 min at 4 °C. Flow cytometry data were recorded on BD LSRII or FortessaX20 using BDFACSDiva (v8.0) software and analysed using FlowJoTM software (v10.4.2, Tree Star).

## Extraction of affinity and functional potency data from published studies

Below, we describe how we obtained the affinity ($K_D$) and functional potency (e.g., $EC_{50}$) values from previous publications. Figure and table numbers refer to those in the original sources.

### 3D affinity data

Alam et al (1996) (Alam et al, 1996): In this publication, $K_D$ values for the OT-I TCR were derived from SPR experiments at 25 °C with either immobilised pMHC complexes or immobilised TCR. The mean $K_D$ values estimated using kinetic parameters $k_{on}$ and $k_{on}$ were provided in Table 1. Sample sizes: OVA = 5, E1 = 2, V-OVA = 3, R4 = 3.

Alam et al (1999) (Alam et al, 1999): Mean $K_D$ values for OT-I TCR were taken from Table 1, measured by SPR at 6 °C, 25 °C, and 37 °C ($N \geq 3$). Only values from 25 °C and 37 °C were used. Biphasic binding behaviour was reported for N4 (named OVA) and A2 at 37 °C, with two $K_D$ values being provided. These were excluded from our analysis.

Rosette et al (2001) (Rosette et al, 2001): We used $K_D$ values from Table 1, which reported OT-I TCR affinity for OVA and G4, measured by SPR at 25 °C and 37 °C. $N$ was not specified. As in Alam et al, (1999), biphasic binding was observed at 37 °C for OVA, and these data were excluded.

Stepanek et al (2014) (Stepanek et al, 2014): In this publication, $K_D$ values for OT-I TCR at 25° was determined using equilibrium binding analysis. The mean $K_D$ values from two independent replicated were provided in Supplementary Fig. S1D (Stepanek et al, 2014).

### 2D affinity data

Huang et al (2010) (Huang et al, 2010): 2D affinity values were measured using the adhesion frequency assay. In this method, naive

CD8$^+$ T cells from OT-I transgenic mice were repeatedly brought into contact with red blood cells presenting pMHC. We extracted mean 2D affinity values (A$_c$K$_a$) from Table 1. Sample size was not reported. We converted A$_c$K$_a$ values to 2D $K_D$ values (A$_c$K$_D$) by taking the inverse: A$_c$K$_D$ = 1/A$_c$K$_a$.

### Potency data

Hogquist et al (1995) (Hogquist et al, 1995): Functional responses of OT-I T cells from transgenic mice were measured using a cytotoxicity assay. We extracted the potency as the peptide concentration producing 10% specific lysis ($P_{10}$) from the dose–response curve in Fig. 2. Peptides that did not elicit a measurable response were excluded.

Rosette et al (2001) (Rosette et al, 2001): Functional data were generated with T cells isolated from OT-I transgenic mice. T cells were then stimulated with pMHC complexes immobilised on plates, and upregulation of CD69 was measured after 24 h. We extracted the EC$_{50}$ values from dose–response curves in Fig. 1.

Daniels et al (2006) (Daniels et al, 2006): Pre-selection OT-I double-positive thymocytes were stimulated with peptide-pulsed APCs. Activation was measured by CD69 expression. EC$_{50}$ values, normalised to N4 and corrected for pMHC binding, were taken from Fig. 1A.

Zehn et al (2009) (Zehn et al, 2009) OT-I T cells were stimulated with peptide-pulsed APCs. Potency was measured by intracellular IFN-$\gamma$ production. EC$_{50}$ values, normalised to N4, were taken from Supplementary Fig. 2C.

Huang et al (2010) (Huang et al, 2010): Functional experiments were performed by culturing naive OT-I splenocytes with peptides. After 66 h, cell proliferation was measured. We extracted the EC$_{50}$ values from Fig. 4. Sample size was not reported.

Lo et al (2019) (Lo et al, 2019): CD8$^+$ Jurkat cells expressing the OT-I TCR were stimulated with peptide-pulsed APCs. Functional response was measured by CD69 upregulation. The EC$_{50}$ values were provided in Supplementary Fig. 7C.

Lo et al (2023) (Lo et al, 2023): Naive or anergic mouse OT-I CD8$^+$ T cells were co-cultured with peptide-pulsed splenocytes. Expression of CD69 was measured after 24 h by flow cytometry. We obtained the potency values by plotting the dose–response data in Fig. 4B (using the source data provided by the publication) and fitting the curves with a sigmoidal model in GraphPad Prism 10 to obtain the EC$_{50}$ values ($N = 1$).

## Data analysis

All data fitting and statistical analysis were carried out in GraphPad Prism 10.

### Obtaining antigen potency

The dose–response data from the T-cell activation experiments were fitted with a four-parameter sigmoidal model in GraphPad Prism 10, using Least-Squares regression. The model was defined as:

$$R(x) = E_{\min} + \frac{E_{\max} - E_{\min}}{1 + \left(\frac{EC_{50}}{x}\right)^H} \tag{3}$$

where $x$ represents the peptide concentration used to load the APCs (in µM). The fitted EC$_{50}$ values were used as potency ($P$) values. EC$_{50}$ values exceeding the highest tested peptide concentration were excluded, ensuring that no extrapolated results were used in the final analysis.

### Determination of discrimination power α

We obtained the discrimination power $\alpha$ by fitting a power law in log-space to the log-transformed potency over affinity data:

$$P' = C + \alpha K'_D \tag{4}$$

where $P' = \log_{10}(P)$ and $K_D' = \log_{10}(K_D)$.

### Fitting of the kinetic proofreading model

The log-transformed potency over affinity data was fit to the kinetic proofreading model using the following equation (Pettmann et al, 2021):

$$y = A + N \log_{10}\left(1 + \frac{(k_{on}10^x)}{k_p}\right) \tag{5}$$

where $Y$ is the log-transformed potency, $X$ is the log-transformed affinity, $k_{on}$ is the on-rate, $k_p$ is the proofreading rate, $N$ is the number of steps, and $A$ is the maximum potency ($y$ intercept). Given that on-rates produce only modest variation between pMHCs, we fixed $k_{on}$ to the value measured for the OT-I/N4 interaction (0.13 µM$^{-1}$ s$^{-1}$) (Pettmann et al, 2023).

## Data availability

This study includes no data deposited in public repositories. All data supporting the findings of this study are available within the paper and its Supplementary Information.

The source data of this paper are collected in the following database record: biostudies:S-SCDT-10_1038-S44318-025-00644-5.

## Peer review information

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

## Acknowledgements

We thank Andrew Sewell and David Cole for providing plasmids for OT-I TCR expression and the NIH tetramer facility for providing purified pMHC. We are grateful for the technical support from the Flow Cytometry Facility at the Kennedy Institute of Rheumatology and the SPR Facility at the Sir William Dunn School of Pathology. The work was funded by a Wellcome Trust Senior Fellowship in Basic Biomedical Sciences (207537/Z/17/Z to OD) and by the UKRI Biotechnology and Biological Sciences Research Council (BB/R015651/1 to AG). For the purpose of Open Access, the author has applied a CC BY public copyright licence to any Author Accepted Manuscript version arising from this submission.

## Author contributions

**Anna Huhn**: Conceptualisation; Data curation; Formal analysis; Investigation; Visualisation; Methodology; Writing—original draft; Writing—review and editing. **Mikhail A Kutuzov**: Data curation; Investigation; Methodology; Writing —review and editing. **Keir Maclean**: Investigation. **Lion F K Uhl**: Investigation; Writing—review and editing. **Jagdish M Mahale**: Investigation; Writing— review and editing. **Audrey Gérard**: Supervision; Funding acquisition; Investigation; Writing—review and editing. **P Anton van der Merwe**: Supervision; Investigation; Methodology; Writing—review and editing. **Omer Dushek**: Conceptualisation; Supervision; Funding acquisition; Investigation; Visualisation; Methodology; Writing—original draft; Project administration; Writing—review and editing.

Source data underlying figure panels in this paper may have individual authorship assigned. Where available, figure panel/source data authorship is listed in the following database record: biostudies:S-SCDT-10_1038-S44318-025-00644-5.

## Disclosure and competing interests statement

The authors declare no competing interests.

# Expanded View Figures

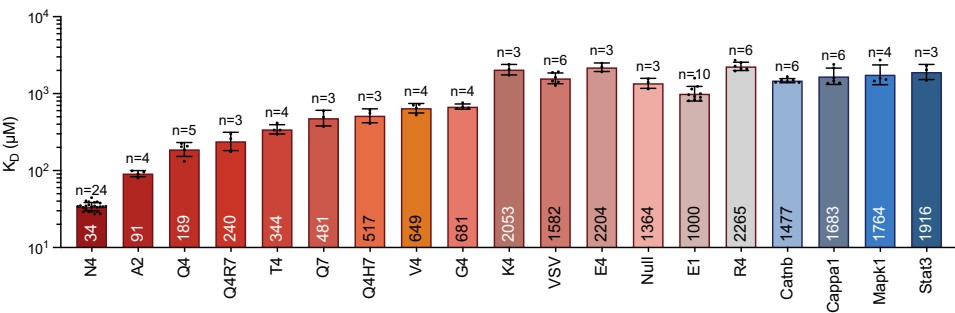

**Figure EV1.  Apparent $K_D$ values for OT-I specific peptides.**

The $K_D$ values were obtained by fitting the steady-state binding using a 1:1 model with $B_{max}$ constrained to value obtained from the B2M or Y3 antibody binding using the standard curve (see Fig. 1 and "Methods"). Geometric mean is displayed within the bars, mean values, error and N are listed in Table 1.

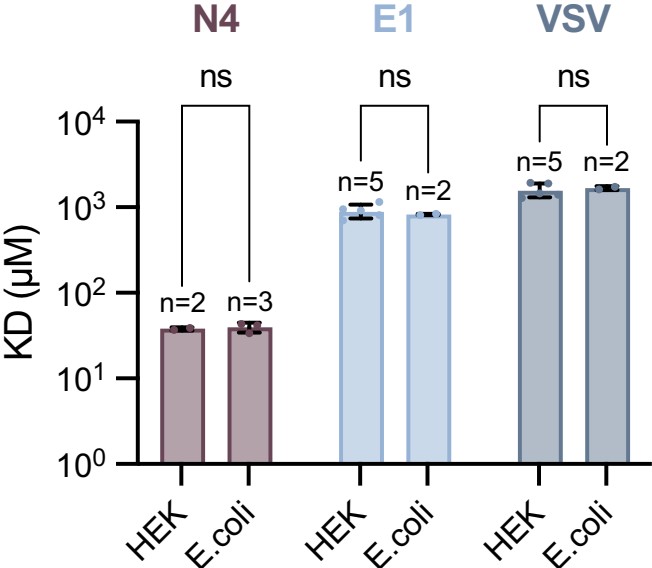

**Figure EV2. The OT-I TCR displays similar affinity to pMHC produced in *E. coli* and HEK293T cells.**

The *E. coli*-produced pMHC were produced in-house, whereas the HEK293T-produced pMHC were supplied by the NIH tetramer facility. Bars show mean values ± SD. Statistical significance was determined using a two-way ANOVA test.

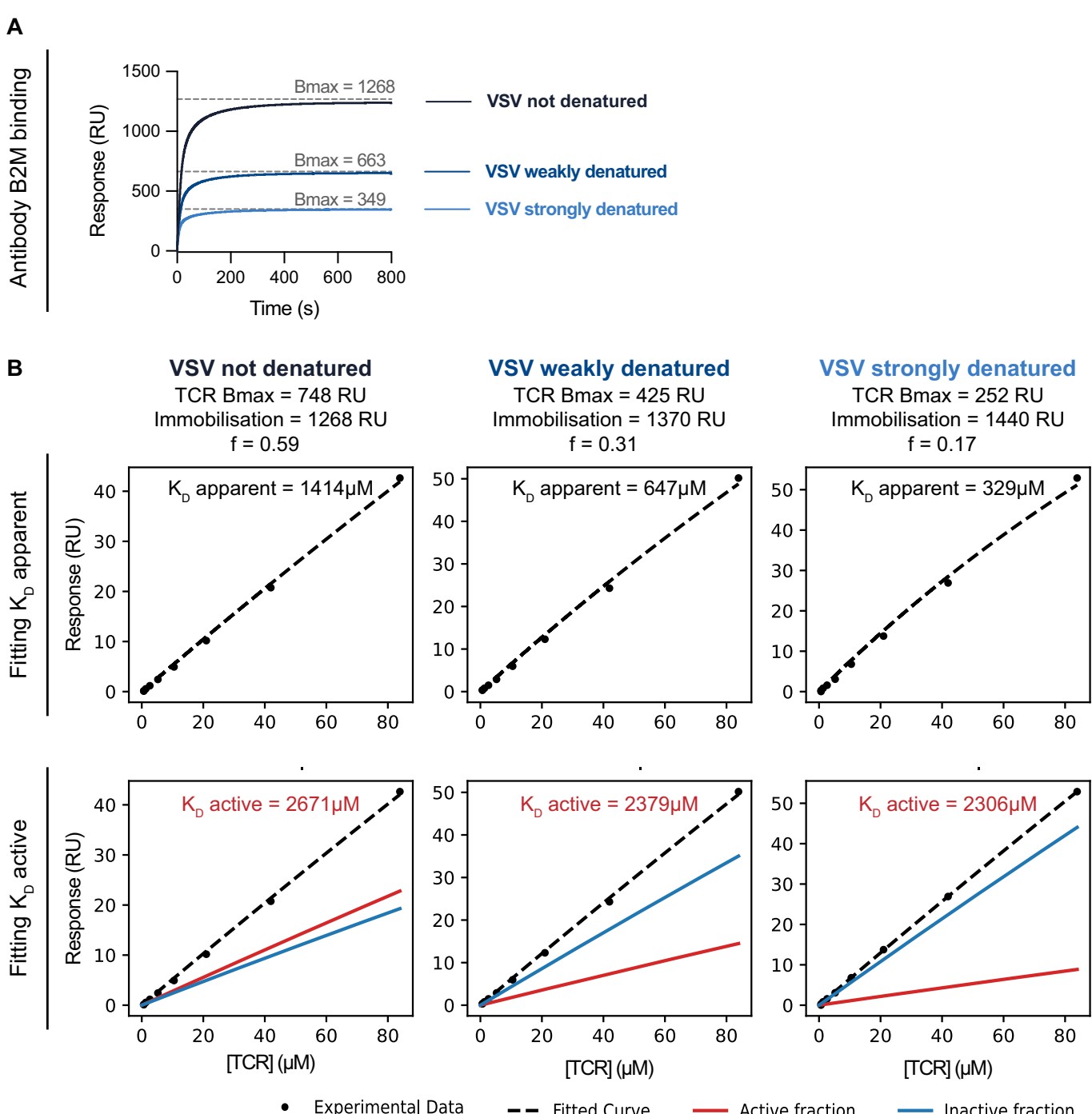

**Figure EV3.** **Calculating the active $K_D$ provides similar results for OT-I binding to surfaces with large differences in the fraction of active pMHC.**

The VSV pMHC was immobilised at similar levels on 3 flow cells in SPR before inducing denaturing by a short (weakly denatured) or long (strongly denatured) injection of glycine solution (pH 1.7). (**A**) B2M antibody binding curves to the three VSV pMHC surfaces. The TCR $B_{max}$ was estimated using the standard curve in Fig. 1B. (**B**) Steady-state TCR binding response to the 3 surfaces (columns). To determine the apparent $K_D$, the data was fit with a 1:1 binding model with constrained $B_{max}$ to determine $K_D$ apparent (top row). To determine the active $K_D$, the workflow in Fig. 2G–I was used, where the fraction of active pMHC was calculated from the ratio of TCR $B_{max}$ to pMHC immobilisation, and $K_D$ inactive was fixed to 2180 μM. While the apparent $K_D$ displayed large differences, the active $K_D$ produced consistent results across all VSV pMHC surfaces.

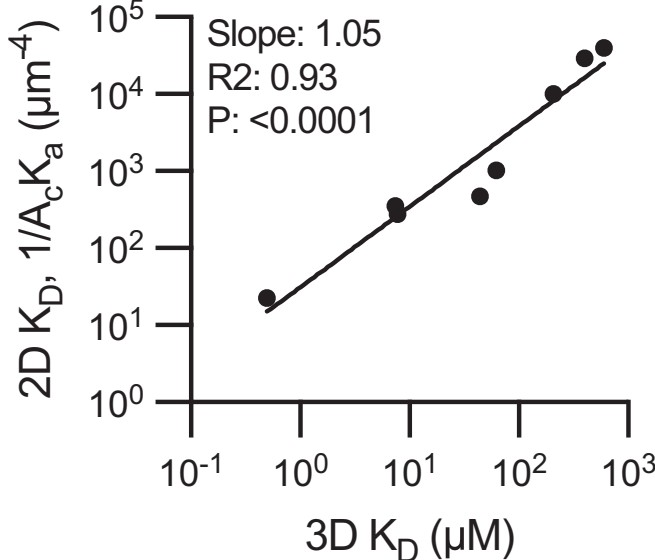

**Figure EV4. Quantitative comparison of 2D and 3D affinities for the 1E6 TCR.**

The log-transformed data was fitted with a linear regression. An *F* test was used to determine a *P* value for the null hypothesis that the slope is equal to zero. All data were taken from Cole et al (Cole et al, 2016).

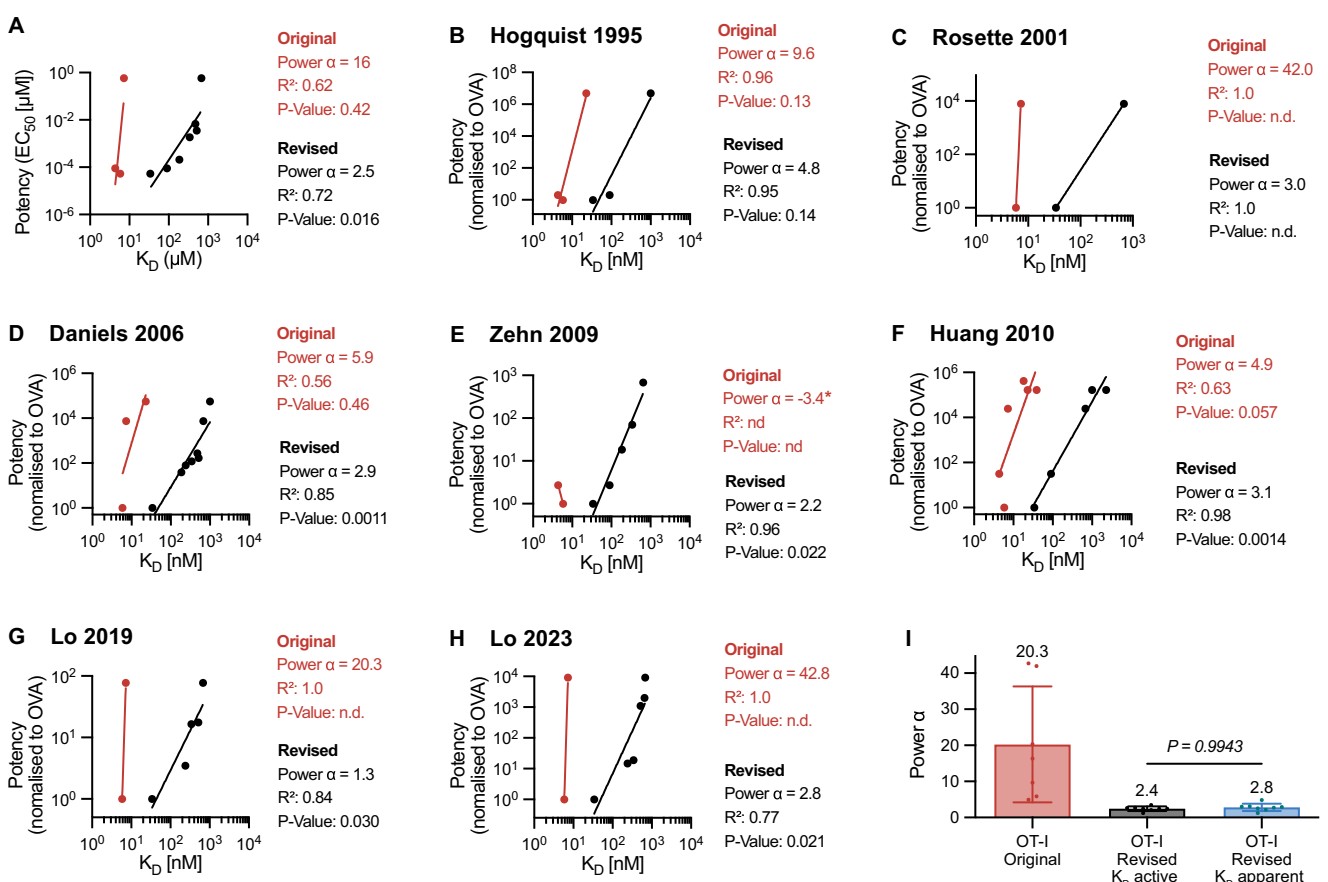

**Figure EV5. Discriminatory power of OT-I TCR calculated with $K_D$ apparent shows imperfect discrimination.**

Plots show peptide potency over original or apparent $K_D$ values. (**A**) Potency data from Fig. 5A. Data is mean of $N = 2$ independent experiments. (**B–H**) Published potency data from the indicated study over original or apparent $K_D$ values. A power law (potency ~ $(K_D)^\alpha$) is fit to the data to estimate the discriminatory power $\alpha$. A Pearson correlation is used to determine $R^2$ and $P$ values on log-transformed values. (**I**) The discriminatory power from (**A–H**) in comparison with discriminatory power calculated with active $K_D$ values. The $P$ value is determined using a t-test. Number of data points included in each category: $N = 7$, $N = 8$, $N = 5$, $N = 15$ for OT-I Original, OT-I revised, other mouse TCRs and other human TCRs, respectively. Bars show mean values (also indicated above the bar) ± SD.

