## [Peer Review File · The EMBO Journal]

Murine T-cell receptor OT-I exhibits imperfect discrimination between foreign and self-antigens

Anna Huhn, Mikhail Kutuzov, Keir Maclean, Lion Uhl, Jagdish Mahale, Audrey Gerard, Philip van der Merwe, and Omer Dushek

Corresponding author(s): Omer Dushek (omer.dushek@path.ox.ac.uk)

Review Timeline:

Submission Date:	9th May 25
Editorial Decision:	25th Jun 25
Revision Received:	14th Aug 25
Editorial Decision:	30th Sep 25
Revision Received:	10th Oct 25
Accepted:	17th Oct 25

Editor: Ioannis Papaioannou

Transaction Report:

Dear Prof. Dushek,

Thank you for submitting your manuscript EMBOJ-2025-121312 for consideration by The EMBO Journal, and for your patience during peer review. Your manuscript has now been seen by three experts in the field, and we have received their informative and constructive reports, which you can find below.

I am pleased to say that, as you will see, the referees indicate interest in your manuscript, and recognize that this is a well-designed and performed work providing novel and significant insights into the regulation of T-cell activation that will be of interest to the community. However, they also identify a number of limitations and raise a few concerns that must be addressed during a thorough revision of the manuscript. The referees provide detailed comments and suggestions for strengthening the manuscript further, and many of their concerns require textual clarification and/or more balanced discussion.

Given the referees' supportive comments and positive recommendations, I would like to invite you to submit a revised version of your manuscript taking the referees' suggestions on board, along with a detailed point-by-point response addressing all referees' comments. I should add that it is The EMBO Journal policy to allow only a single round of major revision, and acceptance of your manuscript will therefore depend on the completeness of your responses in this revised version. Please let me know if you have any questions or comments that you would like to discuss with me. If there are any major points you do not agree with or cannot address during your revision, I would encourage you to share them with me as early as possible to discuss how to proceed further in the most efficient way.

We generally allow three months as standard revision time (September 24, 2025). As a matter of policy, competing manuscripts published during this period will not negatively impact our assessment of the conceptual advance presented by your study. However, we request that you contact us as soon as possible upon publication of any related work, to discuss how to proceed. Should you foresee a problem in meeting this three-month deadline, please let us know in advance and we will be able to grant an extension.

Thank you for the opportunity to consider your work for publication in The EMBO Journal. I look forward to your revision.

Best regards,

Ioannis

Instructions for preparing your revised manuscript

1. When you are ready to submit the revision, please upload:

- A Word file of the manuscript text (including legends of main Figures, EV Figures and Tables). Please make sure that changes are highlighted (or "tracked") to be clearly visible.

- Individual production-quality figure files (one file per figure). When assembling your figures, please refer to our figure preparation guidelines in order to ensure proper formatting and readability in print as well as on screen:

If the data shown in a figure are obtained from n {less than or equal to} 2, please use scatter plots showing the individual data points.

i. the name of the statistical test used to generate error bars and P values

ii. the number (n) of independent experiments (please specify technical or biological replicates) underlying each data point (discussion of statistical methodology can be reported in the Materials and Methods section, but figure legends should contain a basic description of n , P , and the test applied)

iii. the nature of the bars and error bars (s.d., s.e.m.).

- A point-by-point response to the referees' comments, with a detailed description of the changes made (as a word file). All referees' concerns must be fully addressed and their suggestions taken on board. When preparing your letter of response to the referees' comments, please bear in mind that this will form part of the Review Process File and will therefore be available online to the community. Please note that you have the possibility to opt out of the transparent process at any stage prior to publication by letting the editorial office know (contact@embojournal.org); if you do opt out, the Review Process File link will point to the following statement: "No Review Process File is available with this article, as the authors have chosen not to make the review process public in this case.". For more details on our Transparent Editorial Process, please visit our website: <https://www.embopress.org/page/journal/14602075/authorguide#transparentprocess>

- Expanded View (EV) files (replacing Supplementary Information) that are collapsible/expandable online. A maximum of 5 EV Figures can be typeset. EV Figures should be cited as "Figure EV1, Figure EV2" etc. in the text, and their respective legends should be included in the manuscript file after the legends of regular figures. See detailed instructions regarding Expanded View files here: <https://www.embopress.org/page/journal/14602075/authorguide#expandedview>

- For the figures that you do NOT wish to display as Expanded View figures, they should be bundled together with their legends in a single PDF file called "Appendix", which should start with a short Table of Contents (including page numbers). Appendix figures should be referred to in the main text as: "Appendix Figure S1, Appendix Figure S2" etc. Please see detailed instructions here: <https://www.embopress.org/page/journal/14602075/authorguide#expandedview>

- A complete author checklist, which you can download from our author guidelines (<https://www.embopress.org/page/journal/14602075/authorguide>). Please note that the checklist will also be part of the Review Process File.

2. Please note that no statistics should be calculated and shown in Figures if $n=2$. Please also note that each p value should be reported as an exact value.

3. Before submitting your revision, primary datasets (and computer code, where appropriate) produced in this study need to be deposited in appropriate public databases (see <https://www.embopress.org/page/journal/14602075/authorguide#dataavailability>). The accession numbers, database, and the specific URLs (links) should be listed in a formal "Data availability" section (placed after Methods), following the example below:

"The RNA-seq datasets produced in this study are available in the following database:
Gene Expression Omnibus GSE46843 (<https://www.ncbi.nlm.nih.gov/geo/query/acc.cgi?acc=GSE46843>)"

*** All links should resolve to a page where the data can be accessed. ***

*** Please remember to provide in the Data availability section of your revised manuscript reviewer passwords if the datasets are not yet public. ***

*** The Data Availability Section is restricted to new primary data that are part of this study. In case you have no data that require deposition in a public database, please state so instead of referring to the database: "Our study includes no data deposited in public repositories." under the heading "Data availability". ***

4. The materials and methods need to be described in the manuscript using our structured methods format, which is now required for all research articles. According to this format, the Methods section includes a single "Reagents and Tools Table" - listing key reagents, experimental models, software and relevant equipment including their sources and relevant identifiers- followed by a "Methods and Protocols" section describing the methods. Please download and fill our Reagents and Tools Table template (.docx), which you can find in our author guide:

<https://www.embopress.org/page/journal/14602075/authorguide#structuredmethods>. When submitting your revised manuscript, please do not include the Reagents and Tools Table in the Methods section of the manuscript but instead upload it as a separate file choosing the file type "Reagent Table".

5. Please check that the title and the abstract of the manuscript are brief, yet explicit, even to non-specialists. The length of the title should not exceed 100 characters, and the abstract should be a single paragraph not exceeding 175 words.

6. Please also note our reference format: <https://www.embopress.org/page/journal/14602075/authorguide#referencesformat>.

8. Please remember: digital image enhancement is acceptable practice, as long as it accurately represents the original data and conforms to community standards. If a figure has been subjected to significant electronic manipulation, this must be noted in the figure legend or in the "Materials and Methods" section. The editors reserve the right to request original versions of figures and

the original images that were used to assemble the figure.

9. Our journal encourages inclusion of data citations in the reference list to directly cite datasets that were obtained from public databases. Data citations in the article text are distinct from normal bibliographical citations and should directly link to the database records from which the data can be accessed. In the main text, data citations are formatted as follows: "Data ref: Smith et al, 2001" or "Data ref: NCBI Sequence Read Archive PRJNA342805, 2017". In the Reference list, data citations must be labeled with "[DATASET]". A data reference must provide the database name, accession number/identifiers, and a resolvable link to the landing page from which the data can be accessed at the end of the reference. Further instructions are available at: <https://www.embopress.org/page/journal/14602075/authorguide#referencesformat>.

10. We request authors to consider both actual and perceived competing interests. Please review our policy (<https://www.embopress.org/page/journal/14602075/authorguide#conflictsinterest>) and update your competing interests statement if necessary. Please name this section 'Disclosure and competing interests statement' and place it after the Acknowledgements section.

11. Please note that all corresponding authors are required to provide an ORCID ID upon submission of a revised manuscript (<https://orcid.org/>). Please find instructions on how to link your ORCID ID to your account in our manuscript tracking system in our Author guidelines (<https://www.embopress.org/page/journal/14602075/authorguide#authorshipguidelines>).

12. We use CRediT to specify the contributions of each author in the journal submission system. CRediT replaces the author contribution section, which should be removed from the manuscript. Please use the free text box to provide more detailed descriptions. See also guide to authors: <https://www.embopress.org/page/journal/14602075/authorguide#authorshipguidelines>.

14. We would also welcome the submission of cover suggestions or motifs to be used by our Graphics Illustrator in designing a cover.

15. Please use the link below to submit your revision:
<https://emboj.msubmit.net/cgi-bin/main.plex>

Referee #1:

This manuscript, led by Huhn and Dushek, challenges a widely accepted view in the T cell field: that two-dimensional KD affinity measurements more accurately reflect TCR:pMHC interactions than traditional 3D KD values obtained through SPR assays. To address this, the authors undertook a meticulous and technically demanding re-evaluation of the OT-I TCR's affinity for its cognate and variant ligands. Importantly, they performed these measurements at physiological temperature, adding further relevance to their findings. Their analysis revealed that the 3D affinities, when carefully measured, exhibit a graded resolution comparable to that previously observed using 2D methods. They also propose a compelling explanation for the earlier discrepancy: nonspecific binding from inactive or non-signaling TCR complexes likely inflated the apparent affinity in 2D assays. This study represents a critical contribution in refining a foundational model system used across cancer immunotherapy, infectious disease, and basic immunology and implications of this work deserve recognition.

Referee #2:

Since T lymphocytes quantitatively and qualitatively orchestrate the immune response to foreign antigens, thus playing a key role in determining the outcome of infection, it is not surprising that during the past three decades much work was done to unravel the mechanisms of T cell decision-making following their encounter with antigen presenting cells exposing a wide variety of pMHCs.

During the nineties, an important hypothesis was that T cell decision was tightly correlated to the conventional properties of the TCR/pMHC interaction, and particularly the dissociation rate (Matsui et al., Pnas 91:12862, 1994) and affinity constant (Alam et al., 1996, ref 2 of submitted paper). These parameters were dubbed 3D (3 dimensional) since measurements (often based on surface plasmon resonance) involved the interaction between surface-bound molecules and dissolved ligands. This 3D

denomination was based on the implicit, and fairly (not entirely) correct assumption that these parameters provided an accurate description of the interaction between soluble ligands and receptors and that they were intrinsic properties of these receptors.

While aforementioned 3D parameters were found to contribute the activation potency in a number of systematic studies (Aleksic et al.; ref 29), a number of discrepancies remained and it was emphasized in two landmark papers (Huang 2010, ref.20, Huppa 2010 Nature 463:963) that physiological TCR pMHC interactions involved molecules bound to cell surfaces. These were accordingly dubbed 2D interactions. An essential point is that 2D parameters are not well defined: the dissociation rate is dependent on the force exerted by cells on bonds (Huppa et al.). Thus, since TCR engagement was shown to trigger forces (see ref. 53 for fairly delayed forces ; More recently, it was shown that T cells might generate forces of order of 5 pN within a few seconds : Gohring Nature Com 12:2502, 2021). A productive TCR/pMHC interaction might thus generate a pulling force that might break the bond. Interestingly, Huang et al concluded that activating pMHC displayed higher 2D dissociation rate and lower 3D dissociation rate than inactive ones. Also, 2D binding is strongly dependent on intermembrane distance, receptor flexibility and lateral diffusion rates. Accordingly, 2D affinity estimates are dependent on fairly strong assumptions and cannot be considered as intrinsic molecular properties. All this complexity is well illustrated by an important paper (Liu et al. 2014; ref22). In conclusion, it is currently thought that important parameters of TCR activation are affinity, dissociation rate and forces (e.g. Faust J Immunol 211: 333-342, 2023). In addition, it was extensively shown that biomolecule association is a multiphasic process. As a consequence, the dissociation rate and force dependence of a ligand-receptor bond are dependent on its age, as shown for efficient binders such as antibodies (J. Biol. Chem 270:26586, 1995) and even streptavidin (Biophys. J. 89:4374, 2005)

I apologize for this rather lengthy introduction, but I felt it necessary to clarify my opinion concerning the submitted manuscript.

The authors used a clever and innovative method of assessing low affinity interactions that they had previously described (ref. 38) to estimate 3D affinities between OT1 TCR and 20 pMHCs. They convincingly showed that their 3D estimates were better correlated than previous ones to published 2D affinities and activation potencies. Also, they concluded that forces had not a substantial role in their model.

Experiments are well described and conducted. However, my feeling is that the limitations of this study are not sufficiently emphasized.

Specific points:

Title: as explained above, 2D affinities are not well-defined parameters. Also, "perfect" antigen discrimination is not clear: do the authors mean "all-or-none", or do they mean "leading to optimal decision making" (optimal with respect to the efficiency of the immune system)

line 20: dissociation rates are dependent on forces, but surface topography (and microvilli-mediated contacts), intermembrane distance and lateral diffusion strongly influence association rates, and affinity is linked to the ratio between on- and off- rates.

line 25: "some human TCRs" ? or do the authors mean that all human TCRs display weak discrimination ?
Another point is that conclusions might be dependent on the choice of peptides, since the possibility that geometrical parameters might be involved in activation potency is not fully excluded, and the relationship between peptide potency and affinity may be altered by conformational properties (see below comments to lines 75-87).

lines 28-29: is it warranted to compare 2D and 3D parameters, and even to refer to linear correlations between them, since 2D parameters are strongly dependent on studied models.

line 48: "variation" ?

lines 75-87 : while the authors' point seems reasonable, I am not very happy with the implicit assumption that the pMHC may display two conformations: active-correctly folded and inactive. Indeed, dynamic conformation changes in pMHCs are well documented (Yanaka14, vanHateren17 J.Biol.Chem. 292:20255, 2017, Wu19MolCell). The multiplicity of TCR conformations is also well shown (e.g. Fodor18).

Figure 1 - top right plot : it seems that all data points might reasonably fit to a line extrapolating to zero. what is the meaning of the constant values (+57 and -18) ?

Lines 137-148: It is difficult to relate the slope of the correlation between 2D and 3D affinity to the potential importance of forces. Indeed, dissociation rates are known to be dependent of forces (and the dependence was predicted to be exponential by Bell, 1978 Science 200:618) but the relationship between forces and 2D affinity is much less straightforward.

Lines 151-168: the term of "activation potency" of a TCR/pMHC couple is somewhat misleading since T cell activation is dependent on many other parameters, including T cell state, costimulatory signals, cytokine environment. Indeed, even the hierarchy of peptide potency was reported to be dependent on costimulatory signals. (Clement21 Pnas . 118:e2019639118). Also, the dynamics of T cell activation may be different with different peptides (Achar22 Science 376, 880, 2022), and a single EC50 value may not account for all of T cell activation relative to a given TCR-pMHC couple.

Line 203 : what is enhanced discrimination ? do the authors mean "higher than previously estimated" ?

In conclusion: experiments are conducted with state-of-the art methods, and the procedure used to evaluate 3D affinities seems quite promising. However, my feeling is that the limitations of the universality of conclusions might be more cautiously emphasized.

Referee #3:

In this manuscript, Huhn et al. explore the 3D affinities of the OT-I TCR. Prior studies investigating the binding affinity of the OT-I TCR show near perfect discrimination between higher and lower affinity antigens and poor correlations between 3D and 2D affinities. The authors employed a surface plasmon resonance protocol (SPR) developed previously by the team to measure ultra-low TCR/pMHC affinities. Using this protocol, they determined the affinities of the OT-I TCR to 19 pMHC complexes. Through this method, they determine new KD values showing the OT-I TCR has enhanced yet imperfect discrimination. Importantly, these revised 3D affinities now correlate both with 2D affinities and functional responses. The findings of this study clarify important discrepancies between the OT-I TCR and other TCRs and may be important in understanding autoimmunity and peptide specificity in the OT-I model. The results are interesting and potentially provides new insights into the regulation of T cell activation by the strength of TCR-pMHC interactions. However, there are few major concerns regarding the work.

Major comments

1. Huhn et al. show that the 3D K_D values correlate linearly with their 2D counterparts after the SPR protocol correctly accounted for the MHC molecules not presenting the peptides. However, there were several aspects that were unclear for the comparison of the affinities between 3D and 2D. (i) The unbinding rate k_{off} in 2D and 3D binding can be vastly different due to the presence of mechanical forces in 2D (PMID: 24725404). It was unclear by what factors the k_{off} values in the SPR protocol are still different between 2D and the 3D measurements. In the presence of such differences, the proportionality of the KD between 2D and 3D would imply the binding rates k_{on} values are adjusted to compensate for the differences in the k_{off} values. It will be interesting to know how the MHC molecules not presenting the antigen peptides contribute to the k_{on} measurements. (ii) Another possibility is that MHC molecules not presenting the antigen peptides could be present in the 2D measurements and perhaps accounting that in the 2D experiments will make the 2D KD values not correlate in the same manner the report shows. A discussion of these effects would be helpful in parsing out the key take away points for the results reported here.

2. The authors show that the potency (EC₅₀) of the peptides for T cell activation measured by CD69 expression, percentage lysis, or T cell proliferation increases with the estimated KD values here as a power law with a range of exponents $\alpha \approx 0-4$. The theoretical model (McKeithan's kinetic proofreading model) is then employed to glean mechanistic insight in terms of the waiting time and the number of steps in the kinetic proofreading model. However, all the above activation markers involve activation of genes regulated by a multi-faceted signaling kinetics initiated by TCR-pMHC interactions as well as other co-receptor and adhesion receptor interactions. Thus, McKeithan's model which approximates the initial signaling model could be non-trivially affected by downstream signaling and then the gene regulation processes. Therefore, it's unclear if it is reasonable to use this model to relate TCR-pMHC interactions to T cell activation and draw mechanistic insights such as values of α or N within the model. The simplicity of the phenomenological dependence of the activation with the KD is compelling, however, a deeper investigation to relate McKeithan's model to the phenomenological behavior might be required. The potency appears to be evaluated from estimating a peptide concentration that gives rise to 15% of the activation in a cell population assay which contains T cells and target cells containing a large variations of single cell abundances of receptors and other proteins as well as in ligand in target cells. How are those variations incorporated when McKeithan's model is considered? How would the potency vary with KD when a downstream signaling marker such as pErk or Ca⁺⁺ is used to calculate the potency? Is it possible these more upstream signaling markers do not show a simple dependence with KD, however, including the gene regulation somehow compensates for the more complex variations at the signaling scale to produce the simple phenomenological dependency observed here?

Minor comments:

- o Which peptides are represented in Fig 5.? Should the original and revised data have the same n ?
- o It would be interesting to apply this ultralow binding affinity measurement technique to other TCRs such as OT-II.
- o The applications of this research are not immediately clear. A more thorough discussion of future directions could improve this.

Referee 1:

This manuscript, led by Huhn and Dushek, challenges a widely accepted view in the T cell field: that two-dimensional KD affinity measurements more accurately reflect TCR:pMHC interactions than traditional 3D KD values obtained through SPR assays. To address this, the authors undertook a meticulous and technically demanding re-evaluation of the OT-I TCR's affinity for its cognate and variant ligands. Importantly, they performed these measurements at physiological temperature, adding further relevance to their findings. Their analysis revealed that the 3D affinities, when carefully measured, exhibit a graded resolution comparable to that previously observed using 2D methods. They also propose a compelling explanation for the earlier discrepancy: nonspecific binding from inactive or non-signaling TCR complexes likely inflated the apparent affinity in 2D assays. This study represents a critical contribution in refining a foundational model system used across cancer immunotherapy, infectious disease, and basic immunology and implications of this work deserve recognition.

We thank the reviewer for taking the time to read our manuscript and highlighting its impact.

Referee 2:

Since T lymphocytes quantitatively and qualitatively orchestrate the immune response to foreign antigens, thus playing a key role in determining the outcome of infection, it is not surprising that during the past three decades much work was done to unravel the mechanisms of T cell decision-making following their encounter with antigen presenting cells exposing a wide variety of pMHCs.

During the nineties, an important hypothesis was that T cell decision was tightly correlated to the conventional properties of the TCR/pMHC interaction, and particularly the dissociation rate (Matsui et al., Pnas 91:12862, 1994) and affinity constant (Alam et al., 1996, ref 2 of submitted paper). These parameters were dubbed 3D (3 dimensional) since measurements (often based on surface plasmon resonance) involved the interaction between surface-bound molecules and dissolved ligands. This 3D denomination was based on the implicit, and fairly (not entirely) correct assumption that these parameters provided an accurate description of the interaction between soluble ligands and receptors and that they were intrinsic properties of these receptors.

While aforementioned 3D parameters were found to contribute the activation potency in a number of systematic studies (Aleksic et al.; ref 29), a number of discrepancies remained and it was emphasized in two landmark papers (Huang 2010, ref.20, Huppa 2010 Nature 463:963) that physiological TCR pMHC interactions involved molecules bound to cell surfaces. These were accordingly dubbed 2D interactions. An essential point is that 2D parameters are not well defined: the dissociation rate is dependent on the force exerted by cells on bonds (Huppa et al.). Thus, since TCR engagement was shown to trigger forces (see ref. 53 for fairly delayed forces ; More recently, it was shown that T cells might generate forces of order of 5 pN within a few seconds : Gohring Nature Com 12:2502, 2021). A productive TCR/pMHC interaction might thus generate a pulling force that might break the bond. Interestingly, Huang et al concluded that activating pMHC displayed higher 2D dissociation rate and lower 3D dissociation rate than inactive ones. Also, 2D binding is strongly dependent on intermembrane distance, receptor flexibility and lateral diffusion rates. Accordingly, 2D affinity estimates are dependent on fairly strong assumptions and cannot be considered as intrinsic molecular properties. All this complexity is well illustrated by an important paper (Liu et al. 2014; ref22). In conclusion, it is currently thought that important parameters of TCR activation are affinity, dissociation rate and forces (e.g. Faust J Immunol 211: 333-342, 2023). In addition, it was extensively shown that biomolecule association is a multiphasic process. As a consequence, the dissociation rate and force dependence of a ligand-receptor bond are dependent on its age, as shown for efficient binders such as antibodies (J. Biol. Chem 270:26586, 1995) and even streptavidin (Biophys. J. 89:4374, 2005)

I apologize for this rather lengthy introduction, but I felt it necessary to clarify my opinion concerning the submitted manuscript.

The authors used a clever and innovative method of assessing low affinity interactions that they had previously described (ref. 38) to estimate 3D affinities between OT1 TCR and 20 pMHCs. They convincingly showed that their 3D estimates were better correlated than previous ones to published 2D affinities and activation potencies. Also, they concluded that forces had not a substantial role in their model.

Experiments are well described and conducted. However, my feeling is that the limitations of this study are not sufficiently emphasized.

We thank the reviewer for taking the time to read our manuscript and for providing feedback that has improved it.

We have revised our manuscript to include references mentioned in the reviewers introduction

Specific points: Title: as explained above, 2D affinities are not well-defined parameters. Also, "perfect" antigen discrimination is not clear: do the authors mean "all-or-none", or do they mean "leading to optimal decision making" (optimal with respect to the efficiency of the immune system)

We agree that the 2D affinities can be context-dependent and our results are with a specific context that measured the 2D affinities for the OT-I TCR. We could not communicate this nuance within the character limit of the title so have removed it.

Previous studies have defined perfect discrimination as the (apparent) ability of T cells to ignore ligands with just a 3-fold lower affinity even if their concentration increases by 100,000-fold (e.g. (1–3)). This perfect discrimination was largely based on the observation that N4 (WT OVA) can activate OT-I T cells whereas the 3-fold lower-affinity E1 peptide variant could not even when its concentration was increased by 100,000-fold. To quantify this, we previously introduced the discriminatory power (α , the slope of the potency over affinity curve on log-transformed axes) and defined baseline discrimination as $\alpha = 1$, enhanced discrimination as $\alpha > 1$, and perfect discrimination as $\alpha > 9$ (3). Thus, when $1 < \alpha < 9$, as we have found for the OT-I TCR, we refer to it as enhanced but imperfect discrimination. We understand that this terminology is more recent and technical, and therefore have revised the title to remove the term imperfect discrimination.

We have revised the title as follows: The 3D affinities of the OT-I TCR for foreign and self-antigens predict their T cell antigen potency

line 20: dissociation rates are dependent on forces, but surface topography (and microvilli-mediated contacts), intermembrane distance and lateral diffusion strongly influence association rates, and affinity is linked to the ratio between on- and off- rates.

We have revised line 20 to indicate that the 3D and 2D affinities, off-rates, and on-rates can differ by different mechanisms.

line 25: "some human TCRs" ? or do the authors mean that all human TCRs display weak discrimination ? Another point is that conclusions might be dependent on the choice of peptides, since the possibility that geometrical parameters might be involved in activation potency is not fully excluded, and the relationship between peptide potency and affinity may be altered by conformational properties (see below comments to lines 75-87).

We have produced data for the 1G4 and A6 human TCRs and performed a meta-analysis of published studies that included other human and mouse TCRs (3). Collectively, these results were consistent with imperfect antigen discrimination.

We have revised line 25 and the following sentence to make this more explicit.

lines 28-29: is it warranted to compare 2D and 3D parameters, and even to refer to linear correlations between them, since 2D parameters are strongly dependent on studied models.

The original study that reported the 2D affinities (using the adhesion frequency assay with live T cells) argued that the highly non-linear relationship with their 3D affinities explains how T cells can discriminate between small changes in the 3D affinities/off-rates. We now show that this was likely an artefact of inaccurate 3D affinities. Moreover, the observation that 2D and 3D affinities show a linear correlation strongly

suggests that, despite the many factors that might introduce a non-linear relationship between 3D and 2D parameters, T cells can make faithful measurements of the 3D affinity by measuring the 2D affinity.

line 48: "variation" ?

Revised to 'differences'.

lines 75-87 : while the authors' point seems reasonable, I am not very happy with the implicit assumption that the pMHC may display two conformations: active-correctly folded and inactive. Indeed, dynamic conformation changes in pMHCs are well documented (Yanaka14, vanHateren17 J.Biol.Chem. 292:20255, 2017, Wu19MolCell). The multiplicity of TCR conformations is also well shown (e.g. Fodor18).

We have revised the text (and added references) to highlight that pMHC can exist in multiple conformations and that we have simplified these for our purpose.

Figure 1 - top right plot : it seems that all data points might reasonably fit to a line extrapolating to zero. what is the meaning of the constant values (+57 and -18) ?

We agree that the y-intercept of these lines are expected to be zero but as a result of experimental noise/error, we obtained non-zero intercepts. The error estimates on +57 (95% CI:-11.64 to 112.5) and -18 (95% CI:-76.36 to 56.90) overlap with the expected intercept of 0, and we also performed a statistical test (F-Test) to determine a p-value for the null hypothesis that the fitted intercept was equal to 0 finding $p = 0.105$ and $p = 0.7488$, respectively. Therefore, the fitted slopes are not significantly different from zero.

Lines 137-148: It is difficult to relate the slope of the correlation between 2D and 3D affinity to the potential importance of forces. Indeed, dissociation rates are known to be dependent of forces (and the dependence was predicted to be exponential by Bell, 1978 Science 200:618) but the relationship between forces and 2D affinity is much less straightforward.

In Bell's model, the zero-force off-rate (3D off-rate) can increase (slip bond) or decrease (catch bond) by an applied force, and these increases/decreases can have a large non-linear impact on the off-rate under force (since force appears in an exponential factor). If we define the off-rate under force as the "2D" off-rate, then this model suggests that it is possible to have a highly non-linear relationship between 3D and 2D off-rates, and we have discussed this in recent work (4). Given that the 3D and 2D affinities are linearly dependent on the off-rate, it suggests that molecular forces at interfaces can introduce large non-linear relationships between 3D and 2D affinities. We appreciate that a linear relationship between the 3D and 2D affinities on its own cannot be used to determine whether force is acting on the TCR/pMHC bond, or whether processes that act at cellular interfaces but not in solution measurements (e.g. TCR clustering) impact the 2D.

We have revised this paragraph to explain more broadly that 2D affinities can be impacted by any factors/processes that are present at 2D interfaces but not in 3D solution measurements.

Lines 151-168: the term of "activation potency" of a TCR/pMHC couple is somewhat misleading since T cell activation is dependent on many other parameters, including T cell state, costimulatory signals, cytokine environment. Indeed, even the hierarchy of peptide potency was reported to be dependent on costimulatory signals. (Clement21 Pnas . 118:e2019639118). Also, the dynamics of T cell activation may be different with different peptides (Achar22 Science 376, 880, 2022), and a single EC50 value may not account for all of T cell activation relative to a given TCR-pMHC couple.

We apologise for this. Our definition of activation potency is always related to a specific readout of T

cell activation. We do not mean that there exists a universal concentration of antigen that produces 50% activation across all readouts. Indeed, we agree that different T cell subsets in different states will produce different T cell responses depending on the expression of co-signalling receptors and their ligands and the experimental assay (e.g. single end point assay or cumulative readings over time). We defined peptide potency as the concentration of peptide required to elicit a given response (e.g. 50% of the maximum). This generic definition can be applied to any readout at any time since it normalises out the response itself. In other words, the units of peptide potency are the concentration of pulsed peptide rather than the response itself. While different T cell responses do not necessarily correlate well with each other, we and others have found that peptide potency correlates with 3D affinities across many different measures of T cell activation (3). We understand that there are exceptions. Indeed the OT-I TCR was an often cited exception, because the original 3D affinities produced poor correlates of ligand potency. We now show that our revised 3D affinities display high correlations across different measures of T cell activation (Fig 5).

We have revised the main text describing Fig. 5 to explicitly indicate the specific measure of T cell activation used by each study, including target cell lysis, surface receptor up-regulation, proliferation, and cytokine production, and in each case we found high correlations with revised 3D affinities.

Line 203 : what is enhanced discrimination ? do the authors mean "higher than previously estimated" ?

We previously introduced the term discriminatory power or α as the slope of the potency over affinity curve on log-transformed axes (3). We defined baseline discrimination as $\alpha = 1$, enhanced discrimination as $\alpha > 1$, and perfect discrimination as $\alpha > 9$. We found that conventional surface receptors, such as cytokine receptors, G-protein coupled receptors, and receptor tyrosine kinases appear to have baseline discrimination with $\alpha = 1$, which is predicted by a simple occupancy model. However, the T cell receptor exhibited enhanced discrimination with $\alpha \sim 2.0$. This is not consistent with the occupancy model but can be explained by a kinetic proofreading model. Using our revised 3D K_D values, we now find that $\alpha \sim 2.4$ for the OT-I TCR. This shows that this OT-I TCR has enhanced ($\alpha > 1$), but not perfect ($\alpha < 9$), antigen discrimination.

We understand that this terminology is more recent and technical, and therefore have revised this paragraph to explicitly define these categories of discrimination.

In conclusion: experiments are conducted with state-of-the-art methods, and the procedure used to evaluate 3D affinities seems quite promising. However, my feeling is that the limitations of the universality of conclusions might be more cautiously emphasized.

Referee 3:

In this manuscript, Huhn et al. explore the 3D affinities of the OT-I TCR. Prior studies investigating the binding affinity of the OT-I TCR show near perfect discrimination between higher and lower affinity antigens and poor correlations between 3D and 2D affinities. The authors employed a surface plasmon resonance protocol (SPR) developed previously by the team to measure ultra-low TCR/pMHC affinities. Using this protocol, they determined the affinities of the OT-I TCR to 19 pMHC complexes. Through this method, they determine new K_D values showing the OT-I TCR has enhanced yet imperfect discrimination. Importantly, these revised 3D affinities now correlate both with 2D affinities and functional responses. The findings of this study clarify important discrepancies between the OT-I TCR and other TCRs and may be important in understanding autoimmunity and peptide specificity in the OT-I model. The results are interesting and potentially provides new insights into the regulation of T cell activation by the strength of TCR-pMHC interactions. However, there are few major concerns regarding the work.

We thank the reviewer for taking the time to read our manuscript and for providing feedback that has improved it.

Major comments 1. Huhn et al. show that the 3D K_D values correlate linearly with their 2D counterparts after the SPR protocol correctly accounted for the MHC molecules not presenting the peptides. However, there were several aspects that were unclear for the comparison of the affinities between 3D and 2D. (i) The unbinding rate k_{off} in 2D and 3D binding can be vastly different due to the presence of mechanical forces in 2D (PMID: 24725404). It was unclear by what factors the k_{off} values in the SPR protocol are still different between 2D and the 3D measurements. In the presence of such differences, the proportionality of the K_D between 2D and 3D would imply the binding rates k_{on} values are adjusted to compensate for the differences in the k_{off} values. It will be interesting to know how the MHC molecules not presenting the antigen peptides contribute to the k_{on} measurements. (ii) Another possibility is that MHC molecules not presenting the antigen peptides could be present in the 2D measurements and perhaps accounting that in the 2D experiments will make the 2D K_D values not correlate in the same manner the report shows. A discussion of these effects would be helpful in parsing out the key take away points for the results reported here.

We have attempted to measure the kinetics between the OT-I TCR and its various pMHC ligands by SPR. However, these interactions are incredibly fast and we were only able to resolve the OT-I/N4 interaction. We found that this interaction has a relatively normal on-rate but a very fast off-rate that, to our knowledge, is the fastest TCR/pMHC interaction to be measured (off-rates of 5.6 s^{-1} (4)). The technique of SPR, and other biophysical methods (e.g. GCI), are unable to resolve faster off-rates.

We understand the reviewers point that the linear correlation between the 3D and 2D K_D values may hide highly non-linear relationships between the 3D and 2D on-rate and off-rates that perfectly cancel each other out. While we believe that this is unlikely, we agree that we cannot rule this out.

We have introduced this limitation in the discussion.

2. The authors show that the potency (EC_{50}) of the peptides for T cell activation measured by CD69 expression, percentage lysis, or T cell proliferation increases with the estimated K_D values here as a power law with a range of exponents $\alpha \approx 0-4$. The theoretical model (McKeithan's kinetic proofreading model) is then employed to glean mechanistic insight in terms of the waiting time and the number of steps in the kinetic proofreading model. However, all the above activation markers involve activation of genes regulated by a multi-faceted signaling kinetics initiated by TCR-pMHC interactions as well as other co-receptor and adhe-

sion receptor interactions. Thus, McKeithan's model which approximates the initial signaling model could be non-trivially affected by downstream signaling and then the gene regulation processes. Therefore, it's unclear if it is reasonable to use this model to relate TCR-pMHC interactions to T cell activation and draw mechanistic insights such as values of α or N within the model. The simplicity of the phenomenological dependence of the activation with the KD is compelling, however, a deeper investigation to relate McKeithan's model to the phenomenological behavior might be required. The potency appears to be evaluated from estimating a peptide concentration that gives rise to 15% of the activation in a cell population assay which contains T cells and target cells containing a large variations of single cell abundances of receptors and other proteins as well as in ligand in target cells. How are those variations incorporated when McKeithan's model is considered? How would the potency vary with KD when a downstream signaling marker such as pErk or Ca^{++} is used to calculate the potency? Is it possible these more upstream signaling markers do not show a simple dependence with KD, however, including the gene regulation somehow compensates for the more complex variations at the signaling scale to produce the simple phenomenological dependency observed here?

We agree that the kinetic proofreading model is an operational model rather than a molecular model. It does not explicitly include all molecules, their post-translational modifications, and interaction network from the cell surface to changes in gene expression. We note that it is presently not possible to do this without introducing a large number of assumptions and hence error into the model. Nonetheless, the simpler operational proofreading model is useful to explore how a delay between ligand binding and receptor signalling as a result of N steps (taking place with rate k_p) determines the concentration of antigen required to activate T cells when it has a given affinity or off-rate. While we cannot use this method to identify the molecules that mediate the kinetic proofreading steps in T cells, we can use the method to show that a finite number of steps and time-delay are required to explain our data. We and other laboratories (e.g. Jay Groves, Arthur Weiss) are actively trying to identify the molecules that control proofreading.

While we have focused on downstream functional responses in the present work and in our previous work (3), previous studies have examined more proximal signalling readouts (e.g. calcium and a reporter for PLC-g activity) and have arrived at broadly the same conclusion (5, 6). The reason that proximal and distal readouts share a similar level of discrimination is likely a consequence of all of these being downstream of kinetic proofreading. If we define C_i as step i in the proofreading chain, where i goes from 0 to N , then it has previously been shown that ligand discrimination is lowest for small i and is maximal when $i = N$ (the final step) (7, 8). If we assume that the final step is the activation of ZAP70 at the TCR or the formation of LAT condensate, then anything downstream would be expected to display the same level of discrimination.

As the reviewer notes, we have performed our measurements using co-culture of cell populations measuring T cell activation by single-cell flow cytometry. We have plotted the percent of T cells upregulating CD69 over the peptide concentration (e.g. Fig 5A). In this assay, the probability that a T cell will upregulate CD69 at a fixed peptide concentration depends on many factors, including the distribution of peptide on the presenting cells along with variations in the expression of ligands on these cells, the probability that the cell interacts with a T cell, and population-level variations in T cells, including the amount of antigen they require to become activation, which likely depends on molecular abundances (e.g. level of TCR, Lck, etc). These variations determine the steepness (hill number) of the dose-response curve but not the mean ligand potency (e.g. EC50), and this was explored in Fig 1C of Altan-Bonnet & Germain (9). Thus, fitting the mean output from a mathematical model to the mean ligand potency offers a simple way to perform hypothesis testing and determine parameter values. Consistent with this, and despite the large population-level heterogeneities, we have been able to accurately reproduce antigen potency values across experiments including experiments where we knockout molecules that impact kinetic proofreading (10).

We have included a new paragraph in the discussion to explain that similar results have been obtained with proximal and distal readouts of T cell activation, and that work is underway to identify the molecules that contribute to kinetic proofreading in T cells.

Minor comments: o Which peptides are represented in Fig 5.? Should the original and revised data have the same n?

We have performed experiments in Fig 5A/B with the indicated peptides (Fig 5A legend). The rest of the panels use functional data from the indicated published study where we have either used their reported EC50 values or have estimated these from dose-response curves that we have manually digitised from the study. The reason that the number of data points differs between original and revised is because functional data was reported for many peptides whose affinity has not been originally measured.

We have revised the caption to indicate this and included a new Methods section to provide more detail on how the published data was obtained.

o It would be interesting to apply this ultralow binding affinity measurement technique to other TCRs such as OT-II.

We have previously performed a meta-analysis of the discriminatory power of CD4+ T cells and found it to be similar to CD8+ T cells. However, this was performed with published 3D affinities and we have yet to use our method on an MHC-II restricted TCR.

We have revised the discussion to highlight this as a potential application (see also next comment)

o The applications of this research are not immediately clear. A more thorough discussion of future directions could improve this.

While we have focused on the implications of our work, we agree that we could highlight more clearly future applications.

We now include a new discussion paragraph highlighting potential applications of our method including to CD4+ T cells.

References

1. François P, Altan-Bonnet G (2016) The Case for Absolute Ligand Discrimination: Modeling Information Processing and Decision by Immune T Cells. *Journal of Statistical Physics* 162.
2. Ganti RS, et al. (2020) How the T cell signaling network processes information to discriminate between self and agonist ligands. *Proceedings of the National Academy of Sciences* p 202008303.
3. Pettmann J, et al. (2021) The discriminatory power of the t cell receptor. *eLife* 10:1–42.
4. Pettmann J, et al. (2023) Mechanical forces impair antigen discrimination by reducing differences in T-cell receptor/peptide–MHC off-rates. *The EMBO Journal* 42.
5. Yousefi OS, et al. (2019) Optogenetic control shows that kinetic proofreading regulates the activity of the T cell receptor. *eLife* 8:1–33.
6. Tischer DK, Weiner OD (2019) Light-based tuning of ligand half-life supports kinetic proofreading model of T cell signaling. *eLife* 8:1–25.
7. McKeithan TW (1995) Kinetic proofreading in T-cell receptor signal transduction. *Proceedings of the National Academy of Sciences of the United States of America* 92:5042–6.
8. Chan C, George AJ, Stark J (2003) T cell sensitivity and specificity - Kinetic proofreading revisited. *Discrete and Continuous Dynamical Systems - Series B* 3:343–360.
9. Altan-Bonnet G, Germain RN (2005) Modeling T cell antigen discrimination based on feedback control of digital ERK responses. *PLoS biology* 3:e356.
10. Cabezas-Caballero J, et al. (2024) A co-receptor switch reduces T cell cross-reactivity. *bioRxiv preprint*.

Dear Prof. Dushek,

Thank you again for the submission of your revised manuscript (EMBOJ-2025-121312R) to The EMBO Journal for our consideration, and for your patience during peer review. Your manuscript has been sent back to two of the three original referees who had previously assessed the first version of the work, and we have now received their comments, which are included below.

I am very pleased to say that both referees are satisfied with the revision and recommend publication without further comments. In light of this input, I am glad to inform you that your manuscript has been accepted in principle for publication in our journal - congratulations on an excellent work!

There is only one minor textual correction suggested by referee #2 that I kindly request you make in a final version of your manuscript.

From the editorial side, there are also the following few changes we need you to make in this final version of your manuscript, before we can move forward with its formal acceptance and publication in The EMBO Journal:

- Please provide the e-mail address of the corresponding author on the title page of the manuscript, beneath the authors' affiliations.
- Please remove statement "Pre-print server: bioRxiv" from the title page.
- Only one Abstract (a single paragraph) can be present in the revised manuscript (currently, there is an "Abstract" and a "Lay abstract").
- Please move the list of keywords below the Abstract.
- Please move the information provided in the "Open access" statement (NB: it appears twice, on the title page and at the end of Methods) to the "Acknowledgements" section of the revised manuscript (beneath Methods). Please also note that The EMBO Journal is a fully Open Access (OA) journal, and that all articles are published with a CC-BY 4.0 license (please see for more information here: <https://www.embopress.org/open-access>), therefore some information originally included in your "Open Access" statement may be redundant or unnecessary.
- "Funding" should be included in the "Acknowledgements" section of the manuscript.
- Please change heading "Competing interests" to "Disclosure and competing interests statement".
- The author contributions statement should be removed from the manuscript file. Instead, we use CRediT to specify the contributions of each author in the journal submission system. Please feel free to use the free text box to provide more detailed descriptions during submission. See also our guide to authors for more information: <https://www.embopress.org/page/journal/14602075/authorguide#authorshipguidelines>.
- Please include heading "Methods" before mentioning the "Reagents and Tools Table" in the revised manuscript.
- We noticed that there are callouts for the missing Figure S1D and Supplementary Figures 2C and 7C.
- In addition, callouts for figure panels of Fig. EV3-EV4 are missing.
- Please note that the last page of the manuscript Word file is blank.
- The general information table at the top of the Author Checklist (author name, journal, manuscript ID) must be completed.
- The title page of the Appendix PDF file should contain the heading "Appendix for:" followed by the manuscript's title, and a brief Table of Contents including page numbers of the listed items.
- Please note that EMBO press papers are accompanied online by:
 - A) a short (2 sentences) summary of the findings and their significance,
 - B) 2-5 short bullet points highlighting the key results, and
 - C) a synopsis image in .jpg or .png format that is exactly 550 pixels wide and 300-600 pixels high (the height is variable). Please note that all text needs to be legible at the final size.Please upload this information along with your revised manuscript (the text for A and B should be provided in a separate Word file).

- During our routine data checks, our data editors have raised the following queries regarding figures, data, and legends. Please make sure that all requests below are completely addressed in the final version of your manuscript (please highlight all changes in the revised manuscript):

1. Please provide the exact p values in the legend of Figure 5J.
2. Please indicate the statistical test used for data analysis in the legend of Figure EV2.
3. Please note that information related to "n" is missing in the legends of Figures 3A, 6B, EV2, EV4 I.
4. Please note that the error bars are not defined in the legends of Figures 3A, 4C, 5J, EV2, EV4 I.

Please also note that as part of the EMBO publications' Transparent Editorial Process, The EMBO Journal publishes online a Peer Review File along with each accepted manuscript. This File will be published in conjunction with your paper and will include the referee reports, your point-by-point response and all pertinent correspondence relating to the manuscript. You can opt out of this by letting the editorial office know (contact@embojournal.org). If you do opt out, the Peer Review File link will point to the following statement: "No Peer Review File is available with this article, as the authors have chosen not to make the review process public in this case."

We look forward to seeing a final version of your manuscript as soon as possible. Please let us know if you have any questions and use this link to submit your revision: <https://emboj.msubmit.net/cgi-bin/main.plex>.

Best regards,

Ioannis

Referee #2:

The corrections should help a number of readers to grasp the issues discussed in the paper. I strongly recommend publication. Minor points : the sentence "a much wider differences" (in introduction) might be corrected ...

Referee #3:

The authors have addressed the concerns.

All editorial and formatting issues were resolved by the authors.

Dear Omer,

Congratulations on an excellent manuscript! I am very pleased to inform you that it has been accepted for publication in The EMBO Journal. Thank you for comprehensively addressing the initially raised referee concerns and editorial requests for changes.

If you have any questions, please do not hesitate to contact the Editorial Office. Thank you for your contribution to The EMBO Journal. Working with you has been a pleasure.

Best regards,

Ioannis
